# Effect of collagen casing on the quality characteristics of fermented sausage

**Xinlei Yan[☉], Le Yang[☉], Yanni Zhang, Wenying Han, Yan Duan[ID]***

College of Food Science and Engineering, Inner Mongolia Agricultural University, Hohhot, Inner Mongolia, China

☉ These authors contributed equally to this work.
* duanyannmg@outlook.com

## Abstract

### Objective

Fermented sausage is popular all over the world for its rich nutrition and unique flavor. Sausage casing is one of the key factors affecting the quality of fermented sausage. However, there is little information involved in this field.

### Methods

In this study, collagen casings were used as a wrapping material, and natural casings (pig casings) were used as a control. The effects of the two types of casings on biogenic amine content and other quality characteristics of fermented sausage were analyzed with increasing the storage time.

### Results

The results showed that with storage time increasing, the hardness and gumminess of fermented sausage in collagen casing (CC) group were higher than those in pig casing (PC) group ($P<0.05$), while the elasticity in CC group was lower than that in PC group ($P<0.05$). In the processing and storage period, there was no significant difference in the type and content of flavor substances between the two groups. More importantly, the contents of tryptamine, putrescine, cadaverine, histamine, tyramine and phenyethylamine in fermented sausage of CC group were lower than those in PC group ($P<0.05$).

### Conclusion

In conclusion, this study revealed that CC could improve the quality characteristics of fermented sausage and reduce the content of biogenic amines in fermented sausage, which provides a theoretical basis for the choice of casings in industrial production in the future.

**Data Availability Statement:** All relevant data are within the paper and its Supporting Information files.

**Funding:** This work was funded by Food Science and Engineering College Science and Technology Plan Project of Inner Mongolia Agricultural

University, China (SPKJ201904) (https://spy.imau.edu.cn/)and Inner Mongolia Natural Science Foundation Program of China (No.2021MS03012) (http://kjt.nmg.gov.cn/). And Yan D received them. The funders had no role in study design, data collection and analysis, decision to publish, or preparation of the manuscript.

**Competing interests:** The authors have declared that no competing interests exist.

## Introduction

Fermented sausage is one of the common fermented meat products. Due to its unique flavor and taste, it is favored by worldwide consumers [1]. It is usually stored and transported at normal temperature and can be eaten directly without cooking. However, many safety risks occur during the processing and storage of fermented sausages, such as oxidative rancidity, pathogenic bacteria and an excessive content of biological amines [2].

Proteins in meat could be converted into biogenic amines under the action of amino acid decarboxylase secreted by microorganisms in the fermentation process [3]. Meanwhile, biogenic amines are not volatile and have high thermal stability. Thus, the biogenic amine content is a common phenomenon in fermented sausages. A certain amount of biogenic amines is indispensable for human beings, but they are potentially toxic to the human body and even cause obvious adverse reactions when an individual ingests excessively. Therefore, the biogenic amine content is regarded as an important indicator for the safety assessment of fermented meat products, and it has been attracting attention from an increasing number of related research groups [4].

Casings play an important role in the industrial production of fermented sausage [5]. There are a variety of casings with different functional characteristics in market at present according to the different demands for meat packaging material in different countries and regions. Casings can be further divided into natural casings and collagen casings in terms of edibility. Natural casings are made from the fresh intestines of healthy animals after some processing, such as pig and sheep [9]. But it cannot be adapted to large-scale industrialized mass production due to their single source and poor processing performance. Collagen casing is an edible artificial casing, which is made from animal dermis. It has stable performance during industrial filling, which can produce products of uniform specifications, reduce the generation of defective products and achieve maximum benefits [6–10].

The maturation process of fermented sausage is easily affected by the type of sausage casing. At present, most studies focus on the comparison of structural properties between collagen casings and natural casings [11, 12]. However, there is little information about the effects of collagen casing on fermented sausage quality characteristics [2, 5, 13]. Serio *et al.* showed that different types of casings had a great influence on fermented sausages quality, which may affect biogenic amine content and the production of some volatile flavor compounds through affecting proteolysis in ripening time [6]. Škaljac *et al.* indicated that at the end of drying as well as at the end of the storage period, the total content of 13 US-EPA PAHs (U.S. Environmental Protection Agency Polycyclic Aromatic Hydrocarbons) was lower in sausages with collagen casings than in sausages with natural casings [14]. Zając *et al.* showed that sausages in the collagen casing were softer and had higher a* and b* color parameters than those with ovine casings [15].

In this study, two types casings, collagen casings (CC) and natural casings (pig casings, PC), were investigated, and the biogenic amine contents and other quality characteristics of fermented sausage were determined. Our results may provide theoretical evidence for the production of fermented sausage and the selection of its casing types (collagen casing or natural casing) from a security perspective.

## Materials and methods

### Preparation of the fermented sausages

All meat was taken from Sunite sheep that was purchased from Xilin Gol League, Inner Mongolia, China and ingredients were purchased from a supermarket (East tile kiln, Saihan

District, Hohhot, Inner Mongolia, China). The ingredients in traditional formulation (salt 2.5%, Inner Mongolia Ejinur Salt Co., Ltd; sodium nitrate 100 mg/kg; sodium nitrite 70 mg/kg, Sichuan Jinshan Pharmaceutical Co., Ltd, Sichuan, China; glucose 0.5%, Jining Huipeng Chemical Co., Ltd, Jining, Shandong, China; black pepper 0.5% and sucrose 0.5%, Shandong Tuohao Food Co., Ltd, Dezhou, Shandong, China) were added to a mixture of 80% chopped sheep hind leg and 20% sheep tail fat. All fermented sausages were inoculated with a recommended commercial starter culture composed of *Pediococcus pentosaceus* and *Staphylococcus xylosus* (F-1 BactofermTM, Chr-Hansen, Denmark) for traditionally fermented meat products, with an amount of 25 g/200 kg added. The production of fermented sausage was divided into two groups according to the experimental design. The fermented sausages in one group were made by pig casing as packing material (PC group, casing diameter was 22 mm which were purchased from Shunping Qianyuan Electronic Commerce Co., Ltd). And the other group was made by collagen casing as the packing material (CC group, casing diameter was 22 mm, Shunping Qianyuan Electronic Commerce Co., Ltd, Baoding, Hebei, China).

Mutton was chopped up by chopping machine (ZB-25, Zhucheng Holy land food packaging machinery Factory, Weifang, Shandong, China, chopping speed, 1800 rpm; inject rate, 200 rpm) and salted with auxiliary materials (1 day). After curing, fermented sausages were formed by sausage enema mechanism (JQC-II, Zhucheng Holy land food packaging machinery factory, Weifang, Shandong, China, filling speed, 100–500 kg/h and error was less than 5 grams). The stuffed sausages with a diameter of 10 cm were hung in a box with constant temperature and humidity (RH). Fermentation was performed at 25˚C and 95% relative humidity for the first 3 days. For the ripening process, the temperature was set at 16˚C and 90%~70% RH for 14 days. Samples were taken on day 0 (after salting), day 3 (after fermentation), day 11 (after processing), day 21 (after storage for 10 d), day 41 (after storage for 30 d) and day 61 (after storage for 50 d). The parallel test was carried out 3 times (3 batches of sausages were made at the same time), and 3 sausages were randomly selected from each batch at a certain time to determine the relevant indicators.

## Physical and chemical analysis

**Color measurement.**   Color measurement of sausage was carried out with TCP2 automatic color meter (Beijing Xin'ao Yike Photoelectricity Technology Co., Ltd, Beijing, China), which referred to the method of Zheng *et al*. [16]. The sausage products were cut into the slices about 0.5 cm thick and each sample was repeated 3 times individually. The color meter was preheated for 15 min and calibrated with standard plates Y = 87.3, X = 0.3158 and Y = 0.3233 before detection. The instrument according to the sample spot area: φ 20 mm. The standard light source for color measurement was $D_{65}$ light source. The results were expressed in the CLE ($L^*$, luminance value; $a^*$, redness value; and $b^*$, yellowness value) system.

**Water activity analysis.**   The sample was shredded and placed in a vessel (Wuxi Huake Instrument & Meter Co. Ltd, Wuxi, Jiangsu, China), and the water activity was measured using an HD-3A water activity meter (Wuxi Huake Instrument & Meter Co. Ltd, Wuxi, Jiangsu, China). Sample was chopped and putted it into a sample plate (capacity: 20 mL).

**pH value analysis.**   The shredded sample (2 g) was homogenized in 18 ml of normal saline (0.85% NaCl, v/v), and the pH value of the homogenized sample was determined using a PB-10 pH meter (Sartorius, AG, Germany). The pH meter was calibrated with standard solution (pH 4.0 and 7.0).

## Microbiological analysis

The samples were gradually diluted, poured into a plate and were incubated at 37˚C for 24 h in a LRH-150F biochemical incubator (Shanghai Yiheng Technology Co., Ltd, Shanghai, China). The

number of lactic acid bacteria was cultured and counted with MRS (man rogosa sharpe) medium (Nanjing Jianceng Biotechnology Co., Ltd, Nanjing, China, dilution solution: distilled water), and the total aerobic mesophilic bacteria cultured and counted with PCA (plate counting agar) medium (Nanjing Jianceng Biotechnology Co., Ltd, Nanjing, China, dilution solution: distilled water). The appropriate gradient plate (colony count ranged from 30 to 300) was selected for counting, and the average number of colonies on the same dilution plate was calculated multiplied by the corresponding dilution multiple as the colony result per milliliter of sample.

## Texture profile analysis (TPA)

The texture profile was analyzed by a texture analyzer (QTS texture instrument: FTC Corporation, USA). The determination parameters were set as follows: pre-test rate, 2mm/s; pilot test rate, 2mm/s; post-test rate, 2mm/s; test interval, 5s; compression ratio, 50%; trigger force, 5g; trigger type, automatic. The center of the samples was removed and then cut into cubes of 1 cm×1 cm×1 cm. The hardness, elasticity, adhesiveness, gumminess and chewiness were measured by two cycles at room temperature by using a "P-100" probe [17].

## Volatile compound analysis

The volatile compounds of fermented sausages were extracted using HS-SPME and analysed using a GC-MS system (ISQ GC-MS instrument, Thermo Fisher Scientific, USA). The extraction was referred to the method proposed by Luo *et al.* [18] with some modifications. The extraction head (SPME extractor, Supelco, USA) was inserted into the sample flask with a distance of 1 cm from the sample, adsorbed at 60°C for 40 min. And thenthe extraction head was inserted into the injection port (250°C) of the GC instrument (ISQ GC-MS instrument, Thermo Fisher Scientific, USA) to desorb the analytes for 3 min. All samples were extracted in triplicate.

## Determination of biogenic amines

The biogenic amine content was tested by using an Intelligent HPLC (High-performance liquid chromatograph, Agilent Technologies Inc., California, USA) following the previous method report [19]. The determination was performed on C18 column (250mm in length, 4.6mm in inner diameter, packing particle size of 5μm) under the condition of that UV detection wavelength, 254 nm and column temperature, 35°C and the sample size was 20 μL. Mobile phase A was 90% acetonitrile and 10% 0.01mol/L ammonium acetate solution containing 0.1% acetic acid, mobile phase B was 10% acetonitrile and 90% 0.01mol/L ammonium acetate solution containing 0.1% acetic acid. The flow rate of the sample was 0.8 mL/min. The details of gradient elution procedure was shown in Table 1.

The bioamine standard curve was plotted by taking tryptamine, phenylethylamine, putrescine, cadaverine, tyramine, spermidine and histamine standard solution concentrations as the horizontal coordinate and sampling the standard solution concentrations from 0.1, 0.25, 0.5, 1, 1.5, 2.5 and 5 μg/mL in sequence, and the ratio of the obtained peak area to the internal standard peak area as the vertical coordinate.

**Table 1. Liquid phase gradient elution procedure.**

| Time (min) | 0 | 22 | 25 | 32 | 32.01 | 37 |
|---|---|---|---|---|---|---|
| Mobile phase A/% | 60 | 85 | 100 | 100 | 60 | 60 |
| Mobile phase B/% | 40 | 15 | 0 | 0 | 40 | 40 |

## Statistical analysis

The quality data of fermented sausage were summarized using descriptive statistics. The $L^*$, $a^*$, $b^*$, pH and $a_w$ of fermented sausage in the two groups were measured at the processing and storage time points of 0 d, 3 d, 11 d, 21d, 41d and 61d. Biogenic amine content, TPA and volatile flavour composition of fermented sausage in the two groups were measured at the processing and storage time points of 11 d, 21 d, 41 d and 61d. Random terms were included fermentation humidity and ambient temperature. Approximate one-way ANOVA and Tukey's post hoc test for each fixed effect were conducted and critical value for a statistically important effect taken at $P < 0.05$. Statistical analyses were performed using statistics 23.0 software (SPSS, USA). Three technical replicates were measured at each time per treatment, and the data are expressed as the average ± standard deviation (S.D.).

## Results

### Physical and chemical analysis

The results showed that the pH of both the groups decreased first and then increased and reached the lowest value on the third day, indicating the end of fermentation. As shown in Table 2, the pH value of CC group was significantly lower than that of PC group at each period after 3 d of fermentation ($P<0.05$). The $a_w$ values of the two groups remained stable after decreasing but showed no significant difference ($P>0.05$). The values were kept below 0.8 since the end of the dry period (11 d). However, the results showed that $L^*$ in both groups presented the same tendency to level off after decline, and it was significantly higher at 3 d than that at storage period ($P <0.05$). $a^*$ and $b^*$ showed a fluctuating upward trend, and the value at 61 d was highest ($P <0.05$).

### Microorganism analysis

The number of lactic acid bacteria (Table 3) and the total aerobic mesophilic bacteria (Table 4) of fermented sausages in both groups showed a consistent trend over time, which appeared

**Table 2. The effect of the casing type on physical and chemical properties of fermented sausage.**

| Group | | Processing and storage time | | | | | |
|---|---|---|---|---|---|---|---|
| | | 0 d | 3 d | 11 d | 21 d | 41 d | 61 d |
| $L^*$ | PC | 44.3±0.3Da | 44.0±0.3Ca | 36.0±0.6Aa | 36.1±0.4Aa | 36.0±0.08Aa | 39.3±0.6Ba |
| | CC | 44.2±0.07Ca | 43.1±0.4Ba | 34.0±0.1Aa | 35.0±0.5Aa | 34.6±0.3Ab | 34.0±0.3Ab |
| $a^*$ | PC | 20.6±0.2Bab | 20.1±0.2Aa | 24.3±0.07Ca | 21.4±1.2Ca | 21.3±0.2BCa | 24.5±0.04CDa |
| | CC | 21.2±0.1ABa | 21.1±0.08Ba | 22.0±0.9Ba | 19.8±0.2Aa | 20.0±0.1Ab | 22.1±0.2Bb |
| $b^*$ | PC | 9.0±0.04Ba | 9.0±0.1ABa | 10.1±0.1Ba | 8.0±0.7Aa | 8.2±0.3Ba | 9.7±0.1Ba |
| | CC | 9.1±0.01Ba | 8.9±0.04Ba | 8.8±0.3Bb | 7.7±0.09Aa | 7.8±0.2Aa | 9.0±0.3Ba |
| pH | PC | 5.9±0.02Aa | 5.1±0.04Ba | 5.8±0.02Aa | 5.7±0.003Aa | 5.7±0.01Aa | 5.9±0.01Aa |
| | CC | 6.0±0.13Aa | 5.0±0.02Ba | 5.1±0.01Bb | 5.4±0.02Cb | 5.5±0.006Db | 5.3±0.02Cb |
| $a_w$ | PC | 0.9±0.001Aa | 0.9±0.003Ba | 0.8±0.003Ca | 0.7±0.002Da | 0.7±0.001Ea | 0.7±0.003Ea |
| | CC | 0.9±0.001Aa | 0.9±0.001Ba | 0.8±0.003Cb | 0.7±0.001Db | 0.8±0.001Ea | 0.7±0.003Fb |

Data in the table were the mean ± S.D. of the three fermented sausage per experimental group of three independent experiments. Different capital letters indicated significant differences over time within the same group ($P<0.05$) and different lowercase letters indicated significant differences between groups at the same time ($P<0.05$), while the same letter indicated no significant difference ($P>0.05$). PC: pig casing, CC: collagen casing; $L^*$: luminance value; $a^*$: redness value; $b^*$: yellowness value; $a_w$: water activity.

**Table 3. The change in lactic acid bacteria number during processing and storage period.**

| Time (d) | 0d | 3d | 11d | 21d | 41d | 61d |
|---|---|---|---|---|---|---|
| PC | 5.962±0.014Aa | 8.081±0.076Da | 8.151±0.041Da | 7.291±0.033Ba | 7.987±0.092Da | 7.751±0.016Ca |
| CC | 6.326±0.032Ab | 7.974±0.048Da | 8.335±0.039Eb | 7.463±0.095Bb | 7.697±0.068Cb | 7.504±0.013Bb |

Data in the table were the mean ± S.D. of three fermented sausage per experimental group of three independent experiments. Different capital letters indicated significant differences over time within the same group ($P<0.05$) and different lowercase letters indicated significant differences between groups at the same time ($P<0.05$), while the same letter indicated no significant difference ($P>0.05$).

from 0 to 3 d after a rapid increase in fluctuations. However, there was no significant difference between CC and PC groups ($P>0.05$).

## Texture index analysis

Texture quality is an important indicator of the chew ability and taste of meat products, which could determine the quality of sausage. The results showed that hardness of fermented sausage tended to increase with increasing storage time, and hardness value of CC group was significantly higher than that of PC group ($P<0.05$). The gumminess in CC group was significantly higher than that in PC group at 11, 41 and 61 d ($P<0.05$), meanwhile the chewiness in CC group was significantly higher than that in PC group ($P<0.05$) (Table 5).

## Volatile flavour composition of sausages

114 volatile flavor compounds were detected in fermented sausages during processing and storage time, including 20 alcohols, 16 aldehydes, 15 acids, 11 esters, 2 phenols, 3 ketone, 39 terpenes and 8 other compounds (Table 6). With the increase of processing and storage time, the flavor substances in fermented sausage gradually decreased. The contents of terpenes in the two groups accounted for about 1/3 of the total contents. The contents of Acetoin (3-hydroxy-2-butanone) in PC group was significantly higher than that in CC group ($P<0.05$) at 11d and 41d.

## Biogenic amine analysis

It showed that the contents of various biogenic amines in CC group were significantly lower than those in PC group during the 41 d storage period ($P<0.05$) and the total bioamine content in CC group was relatively lower than that in PC group at every stages (Table 7). The contents of tryptamine, cadaverine, phenylethylamine, putrescine, histamine and tyramine in CC group were significantly lower than those in PC group ($P<0.05$), while spermidine was significantly higher than that in PC group ($P<0.05$) (Fig 1). During storage, collagen casings could

**Table 4. Changes of viable bacteria number.**

| Time (d) | 0d | 3d | 11d | 21d | 41d | 61d |
|---|---|---|---|---|---|---|
| PC | 6.756±0.196Aa | 8.189±0.114Da | 7.881±0.055Ca | 7.049±0.157ABa | 7.264±0.186Ba | 7.777±0.109Ca |
| CC | 6.366±0.021Aa | 7.817±0.178CDb | 8.079±0.234Da | 7.408±0.111Bb | 7.233±0.167Ba | 7.497±0.287BCa |

Data in the table were the mean ± S.D. of three fermented sausage per experimental group of three independent experiments. Different capital letters indicated significant differences over time within the same group ($P<0.05$) and different lowercase letters indicated significant differences between groups at the same time ($P<0.05$), while the same letter indicated no significant difference ($P>0.05$).

**Table 5. Changes of TPA during processing and storage period.**

| Characteristic index | Group | 11 d | 21 d | 41 d | 61 d |
|---|---|---|---|---|---|
| Hardness/g | PC | 1042.0±270.4Aa | 3378.9±117.4Ba | 2884.2±454.7Ba | 3258.5±573.8Ba |
| | CC | 2668.9±166.9Ab | 2783.1±248.9Aa | 4228.5±124.5Bb | 4531.5±72.1Ba |
| Elasticity/mm | PC | 0.6±0.04Aa | 0.5±0.002Ba | 0.6±0.05Aa | 0.6±0.005ABa |
| | CC | 0.5±0.008ABa | 0.6±0.008ABb | 0.6±0.01Bb | 0.5±0.01Ab |
| Adhesiveness/g.sec | PC | 0.5±0.02Aa | 0.5±0.008Aa | 0.5±0.02Aa | 0.5±0.02Aa |
| | CC | 0.5±0.01Aa | 0.5±0.007Ba | 0.5±0.006ABa | 0.5±0.01Ba |
| Gumminess/g | PC | 911.8±123.2Aa | 1561.6±40.08Aa | 1430.6±205.8Aa | 1585.0±337.8Aa |
| | CC | 1407.6±56.6Ab | 1407.9±167.7Aa | 1999.7±154.6Ba | 2189.6±43.4Ba |
| Chewiness/g·mm | PC | 585.0±100.9Aa | 698.9±62.7ABa | 935.0±218.5BABa | 1073.0±88.8Ba |
| | CC | 782.4±16.3Aa | 730.8±63.1Aa | 1153.5±79.1Ba | 1156.3±57.9Ba |

Data in the table were the mean ± S.D. of the three fermented sausage per experimental group of three independent experiments. Different capital letters indicated significant differences over time within the same group ($P<0.05$) and different lowercase letters indicated significant differences between groups at the same time ($P<0.05$), while the same letter indicated no significant difference ($P>0.05$).

reduce the production of biogenic amines compared with pig casings, thus effectively improving the safety of fermented sausages.

## Discussion

This study focused on the effect of CC on biogenic amine contents and other quality indicators during the fermentation of sausage. Since lactic acid bacteria produce acid in fermentation process, the pH value of fermented sausage decreases. When it reaches the lowest point, the growth of the starter strain is inhibited, the acid production capacity decreases, and the pH value slowly increases [20]. The pH value of CC group was lower than that of PC group at all stages after fermentation. This phenomenon may be caused by the different tissue structures of these two casings, leading to differences in the contents of amines and ammonia produced by proteolysis and lipid metabolism [21]. Compared with PC group, the lower pH value could inhibit the growth of putrefying bacteria in CC group, which was conducive to maintain quality of fermented sausage during storage. We found that the different color values of fermented sausage made by collagen casings were lower than those of pig casings. Collagen casings have lower transparency, larger thickness, lower elasticity, worse air permeability and worse binding ability with oxygen than natural casings, as they are made of extracted collagen and ingredients [22]. Therefore, it is necessary to improve the color of fermented sausage by adjusting the tissue structure of the collagen casings in further studies.

Brandy *et al.* revealed that the texture was related to mouthfeel of meat products [23], which was an important factor determining whether meat products that can be accepted by consumers. In our study, CC group exhibited improved texture characteristics of fermented sausages compared with PC group. During storage, the decrease of water activity compacted the structure of the sausage and increases its hardness gradually [24]. Meanwhile, the elasticity of PC group was better than that of CC group, because the permeability of collagen casings was stronger [5]. However, the poor elasticity of collagen casing made the sausages more compact and the small molecular products (oligopeptides and small amino acids) were not easily deteriorated or lost after fermentation, which made the sausage have better gumminess [25]. According to the analysis as mentioned above, the structure of fermented sausage in CC group was more delicate and compact than that in PC group.

**Table 6. Volatile flavour compounds identified and quantified (µg/100g) in fermented sausages by GC–MS during storage.**

| Flavor Compounds | RT | Formula | Contents (µg/100g) | | | | | | | |
|---|---|---|---|---|---|---|---|---|---|---|
| | | | 11d | | 21d | | 41d | | 61d | |
| | | | CC | PC | CC | PC | CC | PC | CC | PC |
| **Alcohols (20)** | | | | | | | | | | |
| Ethanol | 1.34 | $C_2H_6O$ | 0.48±0.12a | 0.45±0.16a | 1.09±0.17 | Nd | Nd | 1.16±0.07 | Nd | Nd |
| 1-Butanol, 3-methyl- | 4.02 | $C_5H_{12}O$ | 0.11±0.03a | 0.12±0.03a | Nd | Nd | Nd | Nd | Nd | Nd |
| 2,3-Butanediol | 5.33 | $C_4H_{10}O_2$ | 1.07±0.33a | 1.48±0.33a | 1.32±0.14a | 1.63±0.53a | 1.38±1.01a | 2.38±0.19a | 1.73±0.38a | 1.57±0.07a |
| 1-Hexanol | 7.71 | $C_6H_{14}O$ | 0.06±0.01a | 0.08±0.02a | Nd | Nd | Nd | Nd | Nd | Nd |
| 2-Heptanol | 8.61 | $C_7H_{16}O$ | 0.04±0.02a | 0.04±0.02a | Nd | Nd | Nd | Nd | Nd | Nd |
| 1,6-Octadien-3-ol, 3,7-dimethyl- | 14.15 | $C_{10}H_{18}O$ | 0.54±0.01a | 0.54±0.14a | 0.92±0.02a | 1±0.26a | 0.95±0.33b | 1.35±0.07a | 0.87±0.14a | 0.66±0.1a |
| endo-Borneol | 16.15 | $C_{10}H_{18}O$ | 0.23±0.01a | 0.2±0.05a | 0.33±0.01a | 0.39±0.14a | 0.46±0.21a | 0.51±0.03a | 0.38±0.07a | 0.3±0.05a |
| Terpinen-4-ol | 16.33 | $C_{10}H_{18}O$ | 0.11±0.01a | 0.12±0.03a | 0.17±0.01a | 0.22±0.09a | Nd | 0.27±0.02 | 0.21±0.03a | 0.16±0.02b |
| α-Terpineol | 16.72 | $C_{10}H_{18}O$ | 0.18±0.02a | 0.2±0.05a | 0.36±0.02a | 0.37±0.11a | 0.41±0.15a | 0.53±0.05a | 0.32±0.04a | 0.28±0.05a |
| Geraniol | 17.97 | $C_{10}H_{18}O$ | 0.1±0.01a | 0.11±0.03a | Nd | Nd | Nd | Nd | Nd | Nd |
| 1-Pentanol | 4.83 | $C_5H_{12}O$ | 0.12±0.02b | 0.12±0.04a | 0.54±0.01a | 0.62±0.15a | 0.46±0.16b | 0.7±0.06a | 0.41±0.06a | 0.31±0.05b |
| R-(-)-1,2-propanediol | 7.23 | $C_3H_8O_2$ | Nd | 0.23±0.21 | Nd | Nd | Nd | Nd | Nd | Nd |
| 2,5-Hexanediol | 9.85 | $C_6H_{14}O_2$ | Nd | 0.03±0.01 | Nd | Nd | Nd | Nd | Nd | Nd |
| Ethanolamine | 2.74 | $C_2H_7NO$ | Nd | Nd | 0.67±0.05a | 0.13±0.03b | 0.27±0.07 | Nd | 0.43±0.07a | 0.39±0.06a |
| Diethylene glycol | 3.42 | $C_4H_{10}O_3$ | Nd | Nd | 6.07±0.11a | 6.41±1.84a | 2.79±0.83b | 5.89±0.47a | 3.03±0.63a | 2.6±0.36a |
| 1-Octen-3-ol | 10.89 | $C_8H_{16}O$ | Nd | Nd | 0.42±0.01 | Nd | Nd | Nd | 0.61±0.11 | Nd |
| Isoborneol | 16.16 | $C_{10}H_{18}O$ | Nd | Nd | 0.33±0.01a | 0.39±0.14a | Nd | Nd | Nd | Nd |
| 3-Methyl-1-butanol | 1.44 | $C_5H_{12}O$ | Nd | Nd | Nd | 1.03±0.14 | Nd | 1.33±0.09 | 0.62±0.09b | 2.1±0.19a |
| 2-Butanol, 3-methyl- | 5.34 | $C_5H_{12}O$ | Nd | Nd | Nd | Nd | 1.3±0.49 | Nd | Nd | Nd |
| 3-Cyclohexen-1-ol, 4-methyl-1-(1-methylethyl)-, (R)- | 16.33 | $C_{10}H_{18}O$ | Nd | Nd | Nd | Nd | 0.24±0.13 | Nd | Nd | Nd |
| **Aldehydes (16)** | | | | | | | | | | |
| Pentanal | 3.17 | $C_5H_{10}O$ | 0.56±0.13a | 0.43±0.12a | 0.81±0.08a | 0.84±0.13a | 0.35±0.12b | 0.54±0.03a | 0.37±0.05a | 0.34±0.01a |
| Hexanal | 5.68 | $C_6H_{12}O$ | 0.42±0.07a | 0.43±0.11a | 5.22±0.4a | 6.27±1.2a | 1.16±0.47b | 4.07±0.31a | 1.87±0.33a | 1.16±0.14b |
| Heptanal | 8.65 | $C_7H_{14}O$ | 0.48±0.04a | 0.43±0.1a | 2.38±0.06a | 2.79±0.67a | 1.03±0.3b | 1.69±0.08a | Nd | Nd |
| Benzaldehyde | 10.56 | $C_7H_6O$ | 0.49±0.11a | 0.55±0.12a | 1.69±0.06a | 1.51±0.49a | 1.56±0.57b | 2.73±0.24a | 1.08±0.14a | 0.91±0.15a |
| Benzeneacetaldehyde | 12.88 | $C_8H_8O$ | 0.12±0.01a | 0.15±0.03a | 0.72±0.01a | 0.93±0.24a | 0.59±0.24a | 0.88±0.11a | 0.62±0.07a | 0.52±0.12a |
| Nonanal | 14.33 | $C_9H_{18}O$ | 1.13±0.29a | 1.21±0.34a | 2.5±0.04b | 3.57±0.71a | 1.6±0.53b | 3.38±0.26a | 3.45±0.81a | 2.93±0.26a |
| 2-Heptenal, (E)- | 10.33 | $C_7H_{12}O$ | Nd | Nd | 0.33±0.02a | 0.42±0.15a | Nd | Nd | Nd | Nd |
| Octanal | 11.58 | $C_8H_{16}O$ | Nd | Nd | 2.92±0.09a | 3.79±0.94a | Nd | Nd | Nd | Nd |
| 2-Octenal, (E)- | 13.16 | $C_8H_{14}O$ | Nd | Nd | 0.5±0.09a | 0.51±0.13a | Nd | Nd | Nd | Nd |
| 2-Nonenal | 15.82 | $C_9H_{16}O$ | Nd | Nd | 0.63±0.05a | 0.69±0.19a | Nd | Nd | Nd | Nd |
| 2-Decenal, (E)- | 18.32 | $C_{10}H_{18}O$ | Nd | Nd | 0.22±0.03a | 0.32±0.09a | Nd | 0.39±0.03 | Nd | Nd |
| Butanal, 3-methyl- | 3.17 | $C_5H_{10}O$ | Nd | Nd | Nd | 0.84±0.13 | Nd | 0.54±0.03 | 0.44±0.15a | 0.3±0.02a |
| Decanal | 16.92 | $C_{10}H_{20}O$ | Nd | Nd | Nd | 0.28±0.1 | Nd | Nd | Nd | 0.14±0.04 |
| Pentadecanal- | 27.11 | $C_{15}H_{30}O$ | Nd | Nd | Nd | 0.38±0.01 | 0.42±0.19 | Nd | 0.36±0.06a | 0.34±0.07a |
| Tetradecanal | 27.11 | $C_{14}H_{28}O$ | Nd | Nd | Nd | 0.38±0.01 | 0.42±0.19a | 0.54±0.06a | 0.36±0.06a | 0.34±0.07a |
| 2-Nonenal, (E)- | 15.82 | $C_9H_{16}O$ | Nd | Nd | Nd | Nd | 0.19±0.06b | 0.43±0.07a | Nd | 0.21±0.06 |
| **Acids (15)** | | | | | | | | | | |
| Ala-Gly | 1.18 | $C_5H_{10}N_2O_3$ | 0.37±0.05a | 0.21±0.16a | Nd | Nd | Nd | Nd | Nd | Nd |
| Malonamic acid | 1.28 | $C_3H_5NO_3$ | 0.31±0.13a | 0.33±0.15a | 0.83±0.07a | 0.63±0.16a | 0.28±0.11b | 0.75±0.07a | 0.44±0.07a | 0.4±0.04a |
| Glycine | 1.62 | $C_2H_5NO_2$ | 1.61±0.62a | 2.48±0.86a | Nd | Nd | Nd | 1.97±0.2 | Nd | Nd |
| Acetic acid | 1.93 | $C_2H_4O_2$ | 5.81±1.79a | 8.53±1.57a | 9.63±0.24a | 10.26±3.68a | 9.37±3.4b | 15.49±1.08a | Nd | 9.35±0.89 |
| Methylmalonic acid | 1.28 | $C_4H_6O_4$ | 5.83±1.40 | Nd | 6.07±0.11a | 6.41±1.84a | 2.79±0.83b | 5.89±0.47a | 3.03±0.63a | 2.6±0.36a |
| Propanoic acid, 2-methyl- | 4.47 | $C_4H_8O_2$ | 0.05±0.02a | 0.08±0.03a | Nd | Nd | Nd | Nd | Nd | Nd |

*(Continued)*

**Table 6.** (*Continued*)

| Flavor Compounds | RT | Formula | Contents (µg/100g) | | | | | | | |
|---|---|---|---|---|---|---|---|---|---|---|
| | | | **11d** | | **21d** | | **41d** | | **61d** | |
| | | | **CC** | **PC** | **CC** | **PC** | **CC** | **PC** | **CC** | **PC** |
| Butanoic acid, 3-methyl- | 6.96 | $C_5H_{10}O_2$ | 0.27±0.09b | 0.5±0.13a | Nd | Nd | Nd | Nd | Nd | 0.34±0.06 |
| Butanoic acid, 2-methyl- | 7.17 | $C_5H_{10}O_2$ | 0.11±0.02a | 0.16±0.04a | Nd | Nd | Nd | Nd | Nd | Nd |
| Hexanoic acid | 10.89 | $C_6H_{12}O_2$ | 0.27±0.01a | 0.28±0.23a | 0.95±0.04a | 0.9±0.34a | 0.54±0.3b | 1.27±0.2a | 0.83±0.14a | 0.66±0.13a |
| Octanoic acid | 16 | $C_8H_{16}O_2$ | 0.43±0.07a | 0.59±0.16a | 1.12±0.05a | 1.71±0.64a | 1.35±0.66b | 3.43±0.28a | 1.61±0.38b | 2.65±0.45a |
| n-Decanoic acid | 20.6 | $C_{10}H_{20}O_2$ | Nd | Nd | 0.33±0.01a | 0.36±0.16a | 0.62±0.29a | 0.53±0.04a | 0.66±0.09a | 0.62±0.14a |
| n-Hexadecanoic acid | 28.36 | $C_{16}H_{32}O_2$ | Nd | Nd | 0.69±0.11 | Nd | Nd | Nd | Nd | Nd |
| Octadecanoic acid | 28.36 | $C_{18}H_{36}O_2$ | Nd | Nd | 0.69±0.11 | Nd | Nd | Nd | Nd | Nd |
| Pentanoic acid | 10.95 | $C_5H_{10}O_2$ | Nd | Nd | Nd | 0.9±0.34 | 0.54±0.3 | Nd | 0.83±0.14 | Nd |
| N-Acetyl-L-alanine | 1.26 | $C_5H_9NO_3$ | Nd | Nd | Nd | Nd | 0.28±0.11b | 0.75±0.07a | Nd | Nd |
| **Esters (11)** | | | | | | | | | | |
| Acetic acid, anhydride with formic acid | 2.04 | $C_3H_4O_3$ | 0.02±0.00a | 0.02±0.01a | 0.1±0.02a | 0.03±0.02b | 0.04±0.01a | 0.05±0.01a | Nd | Nd |
| Propanoic acid, 2-hydroxy-, ethyl ester, (S)- | 5.99 | $C_5H_{10}O_3$ | 0.2±0.02a | 0.16±0.04a | Nd | Nd | Nd | Nd | Nd | Nd |
| 2-Heptanol, acetate | 12.44 | $C_9H_{18}O_2$ | 0.08±0.01a | 0.08±0.02a | Nd | Nd | Nd | Nd | Nd | Nd |
| Octanoic acid, ethyl ester | 16.57 | $C_{10}H_{20}O_2$ | 0.15±0.03a | 0.15±0.05a | Nd | 0.53±0.15 | 0.44±0.16b | 0.9±0.06a | 0.44±0.08a | 0.46±0.01a |
| Bornyl acetate | 18.77 | $C_{12}H_{20}O_2$ | 0.06±0.01a | 0.07±0.02a | Nd | Nd | Nd | Nd | Nd | Nd |
| Acetic acid, oxo-, methyl ester | 2.03 | $C_3H_4O_3$ | Nd | Nd | 0.03±0.005a | Nd | Nd | Nd | Nd | Nd |
| n-Caproic acid vinyl ester | 11.07 | $C_8H_{14}O_2$ | Nd | Nd | 1.75±0.07 | 1.72±0.46 | Nd | Nd | Nd | Nd |
| Octanoic acid, methyl ester | 14.76 | $C_9H_{18}O_2$ | Nd | Nd | Nd | Nd | Nd | 0.34±0.04 | Nd | 0.2±0.02 |
| Butyrolactone | 9.22 | $C_4H_6O_2$ | Nd | Nd | Nd | Nd | Nd | Nd | 0.42±0.07a | 0.32±0.02b |
| Butanoic acid, 4-hydroxy- | 9.22 | $C_4H_8O_3$ | Nd | Nd | Nd | Nd | Nd | Nd | 0.42±0.07a | 0.32±0.02b |
| Decanoic acid, ethyl ester | 21.21 | $C_{12}H_{24}O_2$ | Nd | Nd | Nd | Nd | Nd | Nd | Nd | 0.73±0.95 |
| **Phenols (2)** | | | | | | | | | | |
| Phenol | 3.09 | $C_6H_6O$ | 0.31±0.09a | 0.31±0.08a | Nd | Nd | Nd | Nd | 0.6±0.08a | 0.47±0.05b |
| Eugenol | 20.48 | $C_{10}H_{12}O_2$ | 0.24±0.04a | 0.28±0.06a | 0.38±0.05a | 0.39±0.13a | 0.66±0.27a | 0.83±0.07a | 0.5±0.07a | 0.48±0.07a |
| **Ketones (3)** | | | | | | | | | | |
| Acetoin | 3.43 | $C_4H_8O_2$ | 7.11±3.62b | 13.26±3.46a | 6.07±0.11a | 6.41±1.84a | 2.79±0.83b | 5.89±0.47a | 3.03±0.63a | 2.6±0.36a |
| 2,3-Octanedione | 11.07 | $C_8H_{14}O_2$ | Nd | Nd | 1.75±0.07 | Nd | Nd | Nd | Nd | Nd |
| 5-Hepten-2-one, 6-methyl- | 11.05 | $C_8H_{14}O$ | Nd | Nd | Nd | Nd | Nd | Nd | Nd | 0.4±0.07 |
| **Terpenes (39)** | | | | | | | | | | |
| Cyclobutene, 2-propenylidene- | 4.7 | $C_7H_8$ | 0.09±0.01a | 0.08±0.02a | Nd | Nd | Nd | Nd | Nd | Nd |
| α-Pinene | 9.34 | $C_{10}H_{16}$ | 0.2±0.03a | 0.2±0.06a | Nd | Nd | Nd | Nd | Nd | Nd |
| Camphene | 9.86 | $C_{10}H_{16}$ | 0.22±0.04a | 0.27±0.12a | 0.27±0.05 | Nd | 0.31±0.15a | 0.35±0.04a | 0.25±0.04a | 0.17±0.05b |
| Bicyclo[3.1.1]heptane, 6,6-dimethyl-2-methylene-, (1S)- | 10.69 | $C_{10}H_{16}$ | 0.21±0.05a | 0.67±0.63a | Nd | Nd | 1.35±1.07a | 0.28±0.03ab | Nd | 0.76±0.54 |
| β-Pinene | 10.69 | $C_{10}H_{16}$ | 0.21±0.05a | 0.19±0.06a | Nd | Nd | Nd | Nd | Nd | Nd |
| β-Ocimene | 11.98 | $C_{10}H_{16}$ | 2.07±0.28 | Nd | Nd | Nd | Nd | Nd | Nd | Nd |
| 2-Carene | 13.72 | $C_{10}H_{16}$ | 0.07±0.07a | 0.04±0.01a | Nd | Nd | Nd | Nd | Nd | Nd |
| Limonene | 12.18 | $C_{10}H_{16}$ | 2.96±0.38a | 3.03±0.87a | 2.68±0.03a | 2.78±0.84a | 3.71±1.51a | 2.49±2.13ab | 2.55±0.4a | 1.96±0.16b |
| β-Phellandrene | 12.24 | $C_{10}H_{16}$ | 3.49±0.34a | 3.8±0.96a | 2.7±0.07a | 2.74±0.81a | 3.86±1.56a | 4.43±0.53a | 2.85±0.42a | 2.2±0.19b |
| Bicyclo[3.1.0]hexane, 4-methylene-1-(1-methylethyl)- | 10.69 | $C_{10}H_{16}$ | 0.21±0.05 | Nd | 2.7±0.07a | 2.74±0.81a | 3.86±1.56a | 4.43±0.53a | 2.85±0.42a | 2.2±0.19b |
| Cyclohexene, 4-methylene-1-(1-methylethyl)- | 12.24 | $C_{10}H_{16}$ | 3.49±0.34 | Nd | 2.7±0.07a | 2.74±0.81a | 3.86±1.56a | 4.43±0.53a | Nd | 2.2±0.19 |

(*Continued*)

**Table 6.** (Continued)

| Flavor Compounds | RT | Formula | Contents (μg/100g) | | | | | | | |
|---|---|---|---|---|---|---|---|---|---|---|
| | | | 11d | | 21d | | 41d | | 61d | |
| | | | CC | PC | CC | PC | CC | PC | CC | PC |
| Eucalyptol | 12.28 | $C_{10}H_{18}O$ | 0.25±0.02b | 0.11±0.06a | 0.18±0.01a | 0.14±0.03b | 0.18±0.06a | 0.19±0.01a | 0.14±0.02a | 0.03±0.02b |
| γ-Terpinene | 13 | $C_{10}H_{16}$ | 0.15±0.02a | 0.15±0.04a | Nd | Nd | 0.2±0.08 | Nd | Nd | Nd |
| Cyclohexene, 1-methyl-4-(1-methylethylidene)- | 11.83 | $C_{10}H_{16}$ | 0.07±0.04a | 0.13±0.09ab | Nd | Nd | Nd | Nd | Nd | 1.96±0.16 |
| Camphor | 15.53 | $C_{10}H_{16}O$ | 0.09±0.01a | 0.1±0.02a | Nd | Nd | Nd | Nd | Nd | Nd |
| Anethole | 18.99 | $C_{10}H_{12}O$ | 2.47±0.21a | 2.79±0.69a | 4.46±0.08a | 4.85±1.42a | 5.89±2.35a | 7±0.55a | 4.41±0.64a | 4.03±0.62a |
| Cyclohexene, 4-ethenyl-4-methyl-3-(1-methylethenyl)-1-(1-methylethyl)-, (3R-trans)- | 19.85 | $C_{15}H_{24}$ | 0.04±0.01a | 0.03±0.01a | Nd | Nd | Nd | Nd | Nd | Nd |
| Copaene | 20.82 | $C_{15}H_{24}$ | 0.24±0.04a | 0.27±0.07a | 0.57±0.03a | 0.57±0.17a | 0.68±0.22a | 0.85±0.08a | 0.52±0.08a | 0.47±0.09a |
| Caryophyllene | 21.83 | $C_{15}H_{24}$ | 0.66±0.12a | 0.68±0.16a | 1.59±0.05a | 1.75±0.58a | 2.1±0.73a | 2.5±0.16a | 1.54±0.32a | 1.37±0.22a |
| Humulene | 22.63 | $C_{15}H_{24}$ | 0.05±0.01a | 0.06±0.01a | Nd | Nd | Nd | 0.3±0.03 | Nd | 0.11±0.01 |
| Benzene, 1-(1,5-dimethyl-4-hexenyl)-4-methyl- | 23.18 | $C_{15}H_{22}$ | 0.13±0.02a | 0.15±0.03a | 0.59±0.05a | 0.56±0.16a | 0.66±0.26a | 0.85±0.11a | 0.56±0.1a | 0.5±0.12a |
| 1,3-Cyclohexadiene, 5-(1,5-dimethyl-4-hexenyl)-2-methyl-, [S-(R*,S*)]- | 23.47 | $C_{15}H_{24}$ | 0.5±0.14ab | 0.46±0.38a | 1.49±0.04a | 1.69±0.57a | 2.03±0.73a | 2.88±0.36a | 1.6±0.32a | 1.9±0.5a |
| Bicyclo[3.1.1]hept-2-ene, 2,6-dimethyl-6-(4-methyl-3-pentenyl)- | 23.47 | $C_{15}H_{24}$ | 0.5±0.14ab | 0.21±0.32a | 1.49±0.04a | 1.69±0.57a | 2.03±0.73a | 2.88±0.36a | 1.6±0.32a | 1.9±0.5a |
| β-Curcumene | 23.47 | $C_{15}H_{24}$ | 0.5±0.14a | 0.72±0.12b | 1.49±0.04a | 1.69±0.57a | 2.03±0.73a | 2.88±0.36a | 1.6±0.32a | 1.9±0.5a |
| α-Farnesene | 12.43 | $C_{15}H_{24}$ | 0.19±0.03 | Nd | 0.56±0.03a | 0.54±0.15a | 0.72±0.26a | 0.91±0.1a | 0.72±0.13a | 0.53±0.14a |
| β-Bisabolene | 23.74 | $C_{15}H_{24}$ | 0.12±0.03a | 0.14±0.03a | 0.42±0.02a | 0.39±0.11a | 0.5±0.19a | 0.61±0.07a | 0.64±0.17a | 0.38±0.1b |
| 1H-3a,7-Methanoazulene, octahydro-3,8,8-trimethyl-6-methylene-, [3R-(3α,3aβ,7β,8aα)]- | 24.05 | $C_{15}H_{24}$ | 0.17±0.04a | 0.19±0.03a | Nd | 0.63±0.18 | Nd | Nd | Nd | Nd |
| Cedrene | 24.05 | $C_{15}H_{24}$ | 0.17±0.04a | 0.13±0.1a | Nd | Nd | Nd | Nd | Nd | Nd |
| β-Myrcene | 9.11 | $C_{10}H_{16}$ | Nd | 0.78±0.7 | Nd | Nd | Nd | Nd | Nd | Nd |
| Alloaromadendrene | 23.74 | $C_{15}H_{24}$ | Nd | 0.22±0.04 | Nd | Nd | Nd | Nd | Nd | Nd |
| 1,3-Cyclopentadiene, 5-(1-methylethylidene)- | 7.6 | $C_8H_{10}$ | Nd | Nd | 0.19±0.05a | 0.18±0.01a | 0.26±0.1 | Nd | 0.14±0.02 | Nd |
| 3-Carene | 9.34 | $C_{10}H_{16}$ | Nd | Nd | 0.21±0.01 | Nd | Nd | 0.26±0.03 | Nd | 0.13±0.02 |
| D-Limonene | 12.18 | $C_{10}H_{16}$ | Nd | Nd | 2.68±0.03a | 2.78±0.84a | 3.71±1.51a | 2.49±2.13ab | 2.55±0.4 | Nd |
| Cyclohexene, 1-methyl-4-(1-methylethenyl)-, (S)- | 12.18 | $C_{10}H_{16}$ | Nd | Nd | 2.68±0.03a | 2.78±0.84a | Nd | 2.49±2.13 | 2.55±0.4 | Nd |
| Benzene, 1-methyl-4-(1,2,2-trimethylcyclopentyl)-, (R)- | 23.2 | $C_{15}H_{22}$ | Nd | Nd | 0.59±0.05a | 0.56±0.16a | 0.66±0.26a | 0.85±0.11a | 0.56±0.1a | 0.5±0.12a |
| (E)-β-Famesene | 24.06 | $C_{15}H_{24}$ | Nd | Nd | Nd | Nd | 0.79±0.28a | 0.99±0.11a | Nd | 0.65±0.2 |
| 1,3,5-Cycloheptatriene | 4.66 | $C_7H_8$ | Nd | Nd | Nd | Nd | Nd | 0.24±0.02 | Nd | Nd |
| Styrene | 8.31 | $C_8H_8$ | Nd | Nd | Nd | Nd | Nd | 0.71±0.07 | Nd | 0.41±0.07 |
| 1,3,5,7-Cyclooctatetraene | 8.3 | $C_8H_8$ | Nd | Nd | Nd | Nd | Nd | 0.71±0.07 | Nd | Nd |
| **Others (8)** | | | | | | | | | | |
| Toluene | 4.7 | $C_7H_8$ | 0.09±0.01a | 0.08±0.02a | Nd | Nd | Nd | 0.24±0.02 | Nd | 0.24±0.04 |
| Ethylbenzene | 7.33 | $C_8H_{10}$ | 0.09±0.01a | 0.08±0.01a | Nd | Nd | Nd | Nd | Nd | Nd |
| p-Xylene | 7.61 | $C_8H_{10}$ | 0.27±0.05b | 0.25±0.06a | 0.19±0.05a | 0.18±0.01a | Nd | 0.2±0.05 | 0.14±0.02a | 0.15±0.01a |
| Benzene, 1,3-dimethyl- | 12.11 | $C_8H_{10}$ | 0.27±0.05 | Nd | Nd | 0.18±0.01 | Nd | Nd | Nd | Nd |
| o-Xylene | 7.61 | $C_8H_{10}$ | 0.19±0.16a | 0.12±0.16a | Nd | Nd | Nd | Nd | Nd | Nd |
| o-Cymene | 12.1 | $C_{10}H_{14}$ | 0.32±0.05a | 0.32±0.1a | 0.31±0.01b | 0.38±0.05a | 0.45±0.19a | 0.45±0.05a | 0.31±0.04a | 0.22±0.02b |

(*Continued*)

**Table 6.** (Continued)

| Flavor Compounds | RT | Formula | Contents (µg/100g) | | | | | | | |
|---|---|---|---|---|---|---|---|---|---|---|
| | | | 11d | | 21d | | 41d | | 61d | |
| | | | CC | PC | CC | PC | CC | PC | CC | PC |
| Benzene, 1-methyl-3-(1-methylethyl)- | 13.11 | $C_{10}H_{14}$ | 0.32±0.05 | Nd | Nd | Nd | Nd | Nd | Nd | Nd |
| 2-Methoxy-5-methylphenol | 11.58 | $C_8H_{10}O_2$ | Nd | Nd | Nd | Nd | Nd | Nd | Nd | 3.73±0.34 |
| All contents | | | 50.66 | 50.97 | 93.05 | 98.51 | 75.29 | 112.41 | 58.08 | 68.8 |

The results are expressed as the mean ± S.D. (n = 3). Nd indicates no detection. Different lowercase letters indicated significant differences between groups ($P<0.05$), while the same letter indicated no significant difference ($P>0.05$).

The main sources of volatile flavor components in fermented sausage are from adding spices, and flavor compounds formed by the degradation of protein, fat and carbohydrate by enzymes [26, 27]. The contents and types of terpenes which from spices were more in each group. The main components of pepper flavor are β-cardene, limonene, β-pinene and β-laurene [28, 29]. All these substances were detected in our experimental groups, which were mainly from the black pepper in processed raw materials. And it contributes greatly to the flavor of fermented sausage. In addition, carbohydrate metabolism had a great influence on the flavor formation of fermented sausage, which was mainly completed by LAB [30]. Compounds derived from carbohydrate metabolism such as ethanol, 3-hydroxy-2-butanone and acetic acid were detected in two experimental groups. Compared with collagen casings, the good air permeability of pig casings encourages microorganisms such as lactobacillus to proliferate and metabolize carbohydrates, forming flavor substances such as 3-hydroxy-2-butanone.

**Table 7. Changes of biogenic amine contents.**

| Content, mg/kg | Group | 11 d | 21 d | 41 d | 61 d |
|---|---|---|---|---|---|
| Tryptamine | PC | 2.0±0.04Aa | 3.6±0.08Ba | 13.4±0.10Ca | 15.7±0.33Da |
| | CC | 1.4±0.01Ab | 2.1±0.12Bb | 2.3±0.06Cb | 3.2±0.02Db |
| Phenethylamine | PC | 3.1±0.006Aa | 4.7±0.008Ba | 8.5±0.07Ca | 4.8±0.002Da |
| | CC | 3.8±0.002Ab | 4.7±0.01Ba | 7.2±0.003Cb | 3.5±0.008Db |
| Putrescine | PC | 288.2±0.10Aa | 225.8±0.27Ba | 281.8±2.41Ca | 188.8±0.07Da |
| | CC | 77.9±0.02Ab | 104.5±0.15Bb | 118.5±0.02Cb | 185.4±0.31Db |
| Cadaverine | PC | 142.9±0.51Aa | 139.6±0.16Ba | 304.0±2.78Ca | 403.1±0.04Da |
| | CC | 86.3±0.007Ab | 107.5±0.13Bb | 106.3±0.16Cb | 223.4±0.31Db |
| Histamine | PC | 14.5±0.06Aa | 21.3±0.04Ba | 34.2±0.28Ca | 22.9±0.02Da |
| | CC | 8.7±0.03Ab | 8.2±0.04Bb | 12.0±0.02Cb | 9.6±0.03Db |
| Tyramine | PC | 360.1±0.30Aa | 573.1±0.37Ba | 600.4±5.33Ca | 567.0±0.32Ba |
| | CC | 348.1±0.14Ab | 305.9±1.51Bb | 310.7±1.19Cb | 411.5±0.25Db |
| Spermidine | PC | 3.7±0.007Aa | 3.4±0.01Ba | 4.8±0.06Ca | 3.5±0.004Ba |
| | CC | 4.4±0.04Ab | 4.5±0.07Ab | 4.1±0.03Bb | 6.1±0.01Cb |
| Total Content, mg/kg | PC | 814.5 | 971.5 | 1246.8 | 1205.7 |
| | CC | 530.6 | 537.4 | 561.1 | 842.7 |

The data of different bioamines in the table were the mean ± S.D. and the total content were the mean. The three fermented sausage per experimental group of three independent experiments. Different capital letters indicated significant differences over time within the same group ($P<0.05$) and different lowercase letters indicated significant differences between groups at the same time ($P<0.05$), while the same letter indicated no significant difference ($P>0.05$).

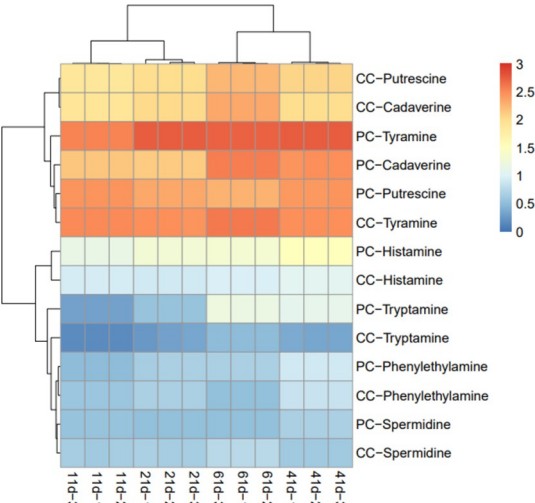

**Fig 1. Biogenic amine content heatmap analysis for fermented sausages during storage.** Each column in heat map represented a different point in time. The ordinate was the relative abundance of each group. The closer the color is to red, the higher the relative abundance of corresponding biogenic amines, and the closer the color is to blue, the lower the relative abundance of corresponding biogenic amines.

We surprisingly found that collagen casings can effectively reduce the contents of biogenic amines in fermented sausage, thus improving the safety of fermented sausage products. During storage, the total contents of biogenic amines in fermented sausage of CC group were all lower than those in PC group. Among them, the content of phenylethylamine in two groups increased firstly and then decreased during storage time. Intermediate storage appeared rise phenomenon because of high temperature during the early stage of fermentation which resulted in stronger activity of phenylalanine decarboxylase and the stronger phenylalanine decarboxylation caused the more accumulation of phenylethylamine finally [31]. Meanwhile, phenylethylamine no longer accumulated in large quantities and gradually reduced by the oxidases produced by microorganisms when decarboxylase activity declined at low temperature storage period (4°C). Otherwise, putrescine is usually used together with cadaverine as the quality indicator of fermented sausage [32]. We thought that the better air permeability of pig casings made the total aerobic mesophilic bacteria in PC group higher than that in CC group in the early stage and then altered the function of the microbial enzyme system that decomposes protein to produce free amino acids. These free amino acids were precursors of biogenic amines, such as ornithine, which resulted in a high content of putrescine [33, 34], which was the main reason for the difference in the biogenic amine content between the two groups [19].

## Conclusions

In this study, two types of casings of fermented sausage were compared and analyzed by the determination of the biogenic amines and quality characteristics. We revealed that collagen casings improved the texture characteristics of fermented sausage, decreased the pH value, and reduced the content of biogenic amines in fermented sausage. However, collagen casing has no special effect on microbial growth and the formation of volatile flavor compounds. In conclusion, this study studied the positive influence of collagen casings on the quality of fermented sausage from the perspective of product safety, and provided a theoretical basis for the application of casings.

## Supporting information

**S1 Table. Data of color measurement during processing and storage of fermented sausages.**
(XLSX)

**S2 Table. Data of water activity measurement during processing and storage of fermented sausages.**
(XLSX)

**S3 Table. Data of pH during processing and storage of fermented sausages.**
(XLSX)

**S4 Table. Data of microbiological indicators during processing and storage of fermented sausages.**
(XLSX)

**S5 Table. Data of TPA measurement during storage of fermented sausages.**
(XLSX)

**S6 Table. Determination results of volatile flavor substances during storage of fermented sausages.**
(XLSX)

**S7 Table. Data of biogenic amine measurement during storage of fermented sausages.**
(XLSX)

**S8 Table. Basic data of heatmap.**
(XLSX)

## Author Contributions

**Conceptualization:** Xinlei Yan, Le Yang.

**Data curation:** Yanni Zhang.

**Formal analysis:** Xinlei Yan, Le Yang.

**Funding acquisition:** Yanni Zhang.

**Methodology:** Yanni Zhang.

**Resources:** Yan Duan.

**Supervision:** Yan Duan.

**Validation:** Yan Duan.

**Visualization:** Xinlei Yan, Le Yang.

**Writing – original draft:** Xinlei Yan, Le Yang.

**Writing – review & editing:** Wenying Han, Yan Duan.

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
