## [Decision Letter · Decision Letter 0]

1 Oct 2021

PONE-D-21-28210Effect of Collagen Casing on the Quality Characteristics of Fermented SausagePLOS ONE

Dear Dr. Duan,

Thank you for submitting your manuscript to PLOS ONE. After careful consideration, we feel that it has merit but does not fully meet PLOS ONE’s publication criteria as it currently stands. Therefore, we invite you to submit a revised version of the manuscript that addresses the points raised during the review process.

ACADEMIC EDITOR: As appended below, the reviewers have raised major concerns/critiques and suggested further justification/work to consolidate the findings. Do go through the comments and amend the MS accordingly. After incorporating all the amendments, the MS should be checked by a native speaker for grammar and syntax errors. 

We look forward to receiving your revised manuscript.

Kind regards,

A. M. Abd El-Aty

Academic Editor

PLOS ONE

Journal Requirements:

3. We understand that you purchased meat, extracts from local markets for this study. In your Methods section, please provide additional regarding the source of this material. Please provide the geographic coordinates and names of the purchase locations (e.g., stores, markets), if available, as well as any further details about the purchased items (e.g., lot number, source origin, description of appearance) to ensure reproducibility of the analyses.

"This work was funded by Food Science and Engineering College Science and Technology Plan Project of Inner Mongolia Agricultural University, China (SPKJ201904) and Inner Mongolia Natural Science Foundation Program of China (No.2021MS03012)."

"This work was funded by Food Science and Engineering College Science and Technology Plan Project of Inner Mongolia Agricultural University, China (SPKJ201904) (https://spy.imau.edu.cn/)and Inner Mongolia Natural Science Foundation Program of China (No.2021MS03012)(http://kjt.nmg.gov.cn/). And Yan D received them. The funders had no role in study design, data collection and analysis, decision to publish, or preparation of the manuscript."

Reviewers' comments:

Reviewer's Responses to Questions

**Comments to the Author**

1. Is the manuscript technically sound, and do the data support the conclusions?

Reviewer #1: Yes

Reviewer #2: Partly

Reviewer #3: Partly

Reviewer #4: Yes

Reviewer #5: Partly

2. Has the statistical analysis been performed appropriately and rigorously? 

Reviewer #1: Yes

Reviewer #2: Yes

Reviewer #3: Yes

Reviewer #4: Yes

Reviewer #5: Yes

3. Have the authors made all data underlying the findings in their manuscript fully available?

Reviewer #1: Yes

Reviewer #2: Yes

Reviewer #3: Yes

Reviewer #4: Yes

Reviewer #5: Yes

4. Is the manuscript presented in an intelligible fashion and written in standard English?

Reviewer #1: Yes

Reviewer #2: No

Reviewer #3: Yes

Reviewer #4: Yes

Reviewer #5: Yes

5. Review Comments to the Author

Reviewer #1: Plos One

PONE-D-21-28210

Effect of Collagen Casing on the Quality Characteristics of Fermented Sausage

Dear Editor,

The article deals with the effect of casing types on the quality characteristics and biogenic amine contents of the fermented sausages. The topic is interesting. It can be published in your journal after necessary corrections have been done. My specific comments and questions are below.

- What was the main purpose of this research? Why was it thought that casing type would affect the formation of biogenic amines? This idea needs to be clearly stated in the introduction section.

- Some information about biogenic amine and health issues and legal limits should be given in the introduction section.

- Line 55: what do you mean “too high”?

- Line 128: Give the incubation conditions!

- Line 131: “total number of colonies” should be “total aerobic mesophilic bacteria” throughout the manuscript.

- Line 141: Much more information about the determination of biogenic amines including the HPLC conditions should be given in the text.

- Fig 1 and Fig 1b should be given as Tables!

- Line 179-180: Please extent the sentence! It is not clear!

- Table 3: Give the total! And discuss it!

- Why the variation in pH values were important but in LAB not?

- Discussion section should be improved!

Reviewer #2: -In the manuscript PONE-D-21-28210, the authors made a comparative analysis of casing effects on sausage quality focusing on biogenic amines and selected physical characteristics. Overall, the limited contents and experiments made the manuscript very facile. Hence, the authors are recommended to evaluate the variations of other bio-molecules, particularly fats, fatty acid, and other common volatile components as well. These molecules are one of the determinants of sausage quality and hence, analyzing the effects of casing could improve the value of the manuscript. The authors should also specify the type of meat used during sausage production, and explicitly describe the HPLC protocol they have followed during the analysis/quantification of the biogenic amines. In addition, the morphology/texture of the sausage samples at each processing and storage time should be supported by optical microscope (SEM) data.

Reviewer #3: Manuscript (PlosOne)

"Effect of collagen casing on the quality characteristics of fermented sausage"

Hereby my comments to the manuscript:

I think you should add an extra keyword that included the color and texture measurements

General remark for the introduction:

Describe what casings are used now for the production of fermented sausages. You do write that there are different types of edible casings, but it would be clearer for the story if you mention that the pig casing is used as a reference, since fermented sausages are made with natural casings.

Line 55: an individual’s intake is too high? -> Do you have numbers? What is too high?

Line 64: I do not agree with the sentence that collagen casings has a similar structure to natural casing! Maybe the collagen type is similar, but how the structures and the collagen network are constructed differ quite a bit. In addition, natural intestine also contains other proteins, such as elastin, while these are not present in the collagen casings. And it is precisely the presence of these other proteins that gives the natural intestine a unique structure and texture characteristics.

Line 79: involved in this study should be replaced by investigated

Line 96: Which type/ brand/ diameter of the collagen casing did you use? Furthermore, with what equipment did you stuff the casings with meat dough? What was the filling pressure, etc? Was this kept the same for the two casing types?

Line 109: How many samples did you measure for the color analysis? What apparatus was used for shredding? Why shredding the samples for the colour analysis. Why didn’t you cut a slice of the sausage and then measure the surface colour?

Line 118: this vessel is specific for the water activity meter?

Line 123: I would at the pH of the saline solution and the concentration/ molarity of the salt solution.

Texture profile analysis: what were the settings of the texture analyser: for example speed, compression % load cell, etc. I think it is good to mention that the gumminess and chewiness of the samples are not measured, but are calculated values.

Line 167: Why is there no results description regarding the measured L, a, b values.

Line 187 – 190: Is a repetition. Already described in material and methods.

Line 213 – 214: Here you already provide an explanation for the results obtained and a conclusion while you do not do that with the other results

Line 234: Do you mean difference in permeability?

Line 240: replace which are to: as they are made

Line 241 -242: How? Are you going to make your own prefabricated collagen casing? Production of a prefabricated collagen casing is a proprietary/ specific process.

Line 243: Brandy et al. -> Brady et al. Furthermore “believed” does not sound very scientific.

Line 245 – 246: With this statement you say that PC is your reference -> is that right? Otherwise you can’t make the statement.

Line 248 – 251: what does the elasticity of the meat stuffed in PC have to do with the permeability of the casing? You are talking about elasticity of the casing, but you did not measure this?

Line 250: collagen casing is not an artificial casing -> artificial casing is a polyamide casing.

Line 251 – 253: Why? And are these small products broken down at PC?

Line 260: permeability for what? Oxygen/ water/ etc.

In general: I would rewrite the discussion part of texture. Now very confusing and not clear.

Reviewer #4: In this study, the effects of collagen casings and natural casings (pig casings) on the biogenic amine content and other quality characteristics of fermented sausage during storage were analyzed. Compared with natural casings, collagen casings could improve the quality characteristics of fermented sausage and reduce the content of biogenic amines in fermented sausage, The study is novel and the results are valuable for the choice of casings in industrial production.

But there are several problems that need to be addressed.

1. How many samples used to measure meat quality traits?

2. In the statistical analysis, “including all interactions between processing and time”, how is that present in the data?

3. There were significant differences in the same group at different periods. So, need to add some discussion about the effect of storage period on biogenic amine content and other quality characteristics of fermented sausage.

4. How collagen casings could reduce the content of biogenic amines in fermented sausage?

5. In the discussion, more theories and references are needed to support the conclusion.

Reviewer #5: The research is important in terms of knowledge about the differences between the sausages in various casings. However, the manuscript cannot be published in a current state. The Discussion is poor and is worth deeper investigation. The conclusion concerning improved texture (harder is better?) is questionable. The statement that CC influenced lower biogenic amines production should be discussed regarding increased levels of BA in PC sausages at the beginning of the fermentation, which means that casings might have contained BA and they were only transferred into the sausage. Taking into account that this is the main conclusion, the manuscript is hard to accept.

Below are my forther detailed comments:

Line 51 why is thermal stability of biogenic amines important if the fermented sausages are usually produced and eaten raw? Do you have in mind any kind of fermented sausages intended to fry/cook?

Line 73 the reference number is missing at Skaliac et al

Line 74 US-EPA PAHs should be explained

Line 75 P<0.05 P should be in italics

Line 76 It should be Zajac et al not Marzena and the number should be at the names as in like 73

Line 125 the space is missing between pH and 4.0

Line 128 The microbiological analysis should be properly described: type of diluting solution, names of microbiological media used, conditions of incubations (time/temperature)

Line 135 the type of texturometer should be mentioned

Line 138 1 cm×1 cm3 3 is missing at first cm

Line 139 the compression conditions should be included

Line 141. The type of chromatograph and detector used should be included. The methods could be described in more detail

Table 1 the significance letters should be at mean values not at errors

Line 170 The information what PC or CC stand for should be added – the table should be self-explanatory

Line 183-190 this information was already given in Materials and methods section

Line 202 There were significant differences between different capital letters in the same group at different periods (P<0.05). There were

significant differences in different lowercase letters among different groups in the

same period (P<0.05). – it should be shortened eg. Capital letters indicate….

Line 234 There were several studies in which fermented sausages produced in collagen casings were produced. Was the same low pH in CC detected? It would be worth discussing

Line 240 This comment would be adequate if the casings production would be a subject of the study. In my opinion it is not suitable here.

Line 243 It should be paraphrased. Tenderness is one of the texture parameters – there is nothing to believe in

Line 245 Why was it improved? Is harder better?

6. PLOS authors have the option to publish the peer review history of their article (what does this mean?). If published, this will include your full peer review and any attached files.

Reviewer #1: No

Reviewer #2: No

Reviewer #3: No

Reviewer #4: No

Reviewer #5: No

---

## [Author Response · Author response to Decision Letter 0]

10 Nov 2021

Dear editor:

Thank you for providing these comments on our submission entitled “Effect of Collagen Casing on the Quality Characteristics of Fermented Sausage”. We appreciate the constructive comments from you and five reviewers. We have studied the comments carefully and provided a detailed response to each point raised.

Response to Editor

Question 1: Please ensure that your manuscript meets PLOS ONE's style requirements, including those for file naming.

Response to question 1: Thank you for your suggestion. We have revised our manuscript followed the style requirement of the journal .

Question 2: We suggest you thoroughly copyedit your manuscript for language usage, spelling, and grammar. If you do not know anyone who can help you do this, you may wish to consider employing a professional scientific editing service.

Response to question 2: Thanks for your helpful suggestions. We sought for the help from AJE (American Journal Experts) to refine our paper, the order number is D3C61K32.

Question 3: We understand that you purchased meat, extracts from local markets for this study. In your Methods section, please provide additional regarding the source of this material. Please provide the geographic coordinates and names of the purchase locations (e.g., stores, markets), if available, as well as any further details about the purchased items (e.g., lot number, source origin, description of appearance) to ensure reproducibility of the analyses.

Response to question 3: Thank you for your careful review. We have added relevant content in our revised manuscript (lines 86-91, 97-101, page 5).

Question 4: We note that you have provided funding information that is not currently declared in your Funding Statement. However, funding information should not appear in the Acknowledgments section or other areas of your manuscript. We will only publish funding information present in the Funding Statement section of the online submission form. Please remove any funding-related text from the manuscript and let us know how you would like to update your Funding Statement.

Response to question 4: Thank you for your suggestion. We were apologized for this error message, we have deleted this part from our manuscript and either from the submission system.

Question 5: Upon re-submitting your revised manuscript, please upload your study’s minimal underlying data set as either Supporting Information files or to a stable, public repository and include the relevant URLs, DOIs, or accession numbers within your revised cover letter.

Response to question 5: All data are in our manuscript.

Question 6: PLOS requires an ORCID iD for the corresponding author in Editorial Manager on papers submitted after December 6th, 2016. Please ensure that you have an ORCID iD and that it is validated in Editorial Manager. 

Response to question 6: Thanks for your suggestion. We have confirmed and uploaded the information on the system.

Response to Reviewer 1

Question 1: What was the main purpose of this research? Why was it thought that casing type would affect the formation of biogenic amines? This idea needs to be clearly stated in the introduction section.

Response to question 1: Thanks for your suggestion. We have added relevant content in our revised manuscript (lines 91-94, 83-86, page 5). 

Question 2: Some information about biogenic amine and health issues and legal limits should be given in the introduction section.

Response to question 2: We searched relevant domestic and foreign laws and regulations. At present, there are no laws and regulations on the content limit standard of other biological amines except histamine. And most of the laws and regulations on histamine focus on fish and fish products. Such as “Zhao JIadi. Screening of bioamine-degrading bacteria and its application in soybean sauce [D]. Jiangnan University, 2020.”

Question 3: Line 55: what do you mean “too high”?

Response to question 3: Thank you for your careful review. We have revised this sentence (lines 64-65, page 4).

Question 4: Line 128: Give the incubation conditions.

Response to question 4: Thanks for your suggestion. We have added relevant content in our revised manuscript (lines 157-158, page 8).

Question 5: Line 131: “total number of colonies” should be “total aerobic mesophilic bacteria” throughout the manuscript.

Response to question 5: Sorry for this mistake. We have corrected the mistake in our revised manuscript (line 161, 223, 315, page 8, 12, 18).

Question 6: Much more information about the determination of biogenic amines including the HPLC conditions should be given in the text.

Response to question 6: Thank you for your constructive suggestion. We have added relevant content in our revised manuscript (lines 180-187, pages 9-10; Table 1).

Question 7: Fig 1 and Fig 1b should be given as Tables!

Response to question 7: Thank you for your constructive suggestion. We have revised these parts in Table.2 and Table.3 of our revised manuscript.

Question 8: Line 179-180: Please extent the sentence! It is not clear!

Response to question 8: Thank you for your careful review. We have modified this sentence of our revised manuscript (lines 223-226, page 12).

Question 9: Table 3: Give the total! And discuss it.

Response to question 9: Thank you for your suggestion. We have added relevant content in our revised manuscript (lines 256-258, 304-305, pages 15-16,18-19).

Question 10: Why the variation in pH values were important but in LAB not?

Response to question 10: Thank you for your careful review. We have revised some contents in our revision (lines 281-283, page 17-18).

Question 11: Discussion section should be improved!

Response to question 11: Thanks for your suggestion. We have improved and added many contents in our revision (Discussion Part).

Response to Reviewer 2

The authors made a comparative analysis of casing effects on sausage quality focusing on biogenic amines and selected physical characteristics. Overall, the limited contents and experiments made the manuscript very facile.

Question 1: The authors are recommended to evaluate the variations of other bio-molecules, particularly fats, fatty acid, and other common volatile components as well. These molecules are one of the determinants of sausage quality and hence, analyzing the effects of casing could improve the value of the manuscript.

Response to question 1: Thanks for your constructive suggestion. We have conducted most of the indexes which you mentioned in our next study to further explore the effect of collagen casings on fat and protein decomposition and flavor of fermented sausages. And the relative article named "The Effect of Collagen Casing on the Flavor Formation of Fermented Sausage" is under modified and prepared to submitted “Meat Science”. We hope you could pay attention to this work if you were interested in our research field.

Question 2: The authors should also specify the type of meat used during sausage production, and explicitly describe the HPLC protocol they have followed during the analysis/quantification of the biogenic amines.

Response to question 2: Thanks for your suggestion. We have added relevant content in our revised manuscript (lines 180-187, pages 9-10; Table 1).

Question 3: The morphology/texture of the sausage samples at each processing and storage time should be supported by optical microscope (SEM) data.

Response to question 3: Thank you for your suggestion. Our study focused on the effect of casings on the quality characteristics of fermented sausage mainly. We will use the optical microscope to conduct in-depth research on the tissue structure of casing in the subsequent experiments.

Response to Reviewer 3

"Effect of collagen casing on the quality characteristics of fermented sausage"

Hereby my comments to the manuscript:

Question 1: I think you should add an extra keyword that included the color and texture measurements.

Response to question 1: Thanks for your suggestion. We have added relevant content in our revised manuscript (line 46, page 3).

Question 2: Describe what casings are used now for the production of fermented sausages. You do write that there are different types of edible casings, but it would be clearer for the story if you mention that the pig casing is used as a reference, since fermented sausages are made with natural casings.

Response to question 2: Thanks for your suggestion. We have revised the paragraph (lines 72-73, page 4).

Question 3: Line 55: an individual’s intake is too high? -> Do you have numbers? What is too high?

Response to question 3: Thank you for your careful review. We have revised this inappropriate statement (lines 64-65, page 4).

Question 4: Line 64: I do not agree with the sentence that collagen casings has a similar structure to natural casing! Maybe the collagen type is similar, but how the structures and the collagen network are constructed differ quite a bit. In addition, natural intestine also contains other proteins, such as elastin, while these are not present in the collagen casings. And it is precisely the presence of these other proteins that gives the natural intestine a unique structure and texture characteristics.

Response to question 4: Sorry for this mistake. We have corrected the mistake in our revised manuscript. (lines 75-76, page 5).

Question 5: Line 79: involved in this study should be replaced by investigated.

Response to question 5: Thank you for your suggestion. We have revised this mistake (line 92, page 5).

Question 6: Line 96: Which type/ brand/ diameter of the collagen casing did you use? Furthermore, with what equipment did you stuff the casings with meat dough? What was the filling pressure, etc? Was this kept the same for the two casing types? Line 96: Which type/ brand/ diameter of the collagen casing did you use? Furthermore, with what equipment did you stuff the casings with meat dough? What was the filling pressure, etc? Was this kept the same for the two casing types?

Response to question 6: Thank you for your careful review. We have added relevant content in our revised manuscript (lines 112-116, pages 6).

Question 7: Line 109: How many samples did you measure for the color analysis? What apparatus was used for shredding? Why shredding the samples for the color analysis. Why didn’t you cut a slice of the sausage and then measure the surface color?

Response to question 7: Sorry for this mistake. We have corrected the mistake in our revised manuscript (lines 134-137, page 7).

Question 8: Line 118: this vessel is specific for the water activity meter?

Response to question 8: Thank you for your careful review. The water activity meter has special vessel. We have added some contents in our manuscript (lines 147-148, page 8).

Question 9: Line 123: I would at the pH of the saline solution and the concentration/ molarity of the salt solution. Texture profile analysis: what were the settings of the texture analyzer: for example speed, compression % load cell, etc. I think it is good to mention that the gumminess and chewiness of the samples are not measured, but are calculated values.

Response to question 9: Thank you for your careful review. We have added some contents in our revision (lines 151-152, 169-172, pages 8, 9). According to the actual situation, the viscosity and chewiness data of the texture tester were obtained directly after the test without calculation.

Question 10: Line 167: Why is there no results description regarding the measured L, a, b values.

Response to question 10: Thanks for your suggestion. We have modified this part (lines 209-213, pages 11).

Question 11: Line 187-190: Is a repetition. Already described in material and methods.

Response to question 11: Thanks for your careful review. We have deleted this part in our revision.

Question 12: Line 213-214: Here you already provide an explanation for the results obtained and a conclusion while you do not do that with the other results

Response to question 12: Thank you for your careful review. We have revised this sentence in our manuscript (lines 293-301, page 17).

Question 13: Line 234: Do you mean difference in permeability? 

Response to question 13: Thank you for your careful review. The structure of different casings makes casings different in air permeability.

Question 14: Line 240: replace which are to: as they are made.

Response to question 14: Thanks for your suggestion. we have corrected this part in our revised manuscript (lines 286-287, page 17).

Question 15: Line 241 -242: How? Are you going to make your own prefabricated collagen casing? Production of a prefabricated collagen casing is a proprietary/ specific process.

Response to question 15: Thank you for your constructive suggestion. We think there are many ways to improve the color of fermented sausage. Such as, we could improve the structure of collagen casing and also could modify the formula of fermented sausage.

Question 16: Line 243: Brandy et al. -> Brady et al. Furthermore “believed” does not sound very scientific.

Response to question 16: Thank you for your careful review. We have corrected this part in our revised manuscript (line 290, page 17).

Question 17: Line 245 – 246: With this statement you say that PC is your reference -> is that right? Otherwise you can’t make the statement.

Response to question 17: Thanks for your suggestion. The main purpose of our research was to study the characteristics of collagen casings and pig casings on fermented sausage quality and pig casings were used as the control group.

Question 18: Line 248 – 251: what does the elasticity of the meat stuffed in PC have to do with the permeability of the casing? You are talking about elasticity of the casing, but you did not measure this?

Response to question 18: Thank you for your careful review. We have corrected this part in Table.5 of our revised manuscript.

Question 19: Line 250: collagen casing is not an artificial casing -> artificial casing is a polyamide casing.

Response to question 19: Thank you for your suggestion. We searched relevant literature, which pointed out that collagen casings are artificial casings. Patricia Suurs, Shai Barbut. Collagen use for co-extruded sausage casings – A review. Trends in Food Science & Technology. 2020; 102: 91-101. doi: 10.1016/j.tifs.2020.06.011

Question 20: Line 251 – 253: Why? And are these small products broken down at PC?

Response to question 20: Thanks for your review. We have added some content in our revised manuscript. According to the following discussion on the content of biogenic amine in fermented sausage could also explain your problem (lines 296-299, 316-319, page 17,18).

Question 21: Line 260: permeability for what? Oxygen/ water/ etc.

Response to question 21: Thank you for your careful review. We have corrected this part in our revised manuscript (lines 285-286, 315, pages 17,18).

Question 22: In general: I would rewrite the discussion part of texture. Now very confusing and not clear.

Response to question 22: Thanks for your suggestion. We have revised the paragraph (lines 293-301, page 17).

Response to Reviewer 4

In this study, the effects of collagen casings and natural casings (pig casings) on the biogenic amine content and other quality characteristics of fermented sausage during storage were analyzed. Compared with natural casings, collagen casings could improve the quality characteristics of fermented sausage and reduce the content of biogenic amines in fermented sausage, The study is novel and the results are valuable for the choice of casings in industrial production.

Question 1: How many samples used to measure meat quality traits?

Response to question 1: Thank you for your careful review. We have added some content in our revised manuscript (lines 127-130, page 7; Table 2, Table 3, Table 4, Table 5, Table 6).

Question 2: In the statistical analysis, “including all interactions between processing and time”, how is that present in the data?

Response to question 2: Thanks for your suggestion. We have revised this paragraph (lines 192-195, page 10).

Question 3: There were significant differences in the same group at different periods. So, need to add some discussion about the effect of storage period on biogenic amine content and other quality characteristics of fermented sausage.

Response to question 3: Thank you for your suggestion. This research focused on the quality characteristics of fermented sausage in different casings. The reason why there is no detailed analysis of the quality of fermented sausage in different periods is to avoid the separation of our main purpose.

Question 4: How collagen casings could reduce the content of biogenic amines in fermented sausage?

Response to question 4: Thank you for your careful review. There are not many relevant literatures to reveal the effect of collagen casings on fermented sausage bioamines. The starting point for our research was to explore whether collagen casing has an effect on the bioamine production of fermented sausage. The results showed that collagen casings can effectively reduce biogenic amine production, and this phenomenon was discussed and analyzed in discussion part (lines 304-313, page 17-18).

Question 5: In the discussion, more theories and references are needed to support the conclusion.

Response to question 5: Thanks for your suggestion. We have improved and added many contents in our revision (Discussion Part).

Response to Reviewer 5

The research is important in terms of knowledge about the differences between the sausages in various casings. However, the manuscript cannot be published in a current state. The Discussion is poor and is worth deeper investigation. The conclusion concerning improved texture (harder is better?) is questionable. The statement that CC influenced lower biogenic amines production should be discussed regarding increased levels of BA in PC sausages at the beginning of the fermentation, which means that casings might have contained BA and they were only transferred into the sausage.

Question 1: Line 51 why is thermal stability of biogenic amines important if the fermented sausages are usually produced and eaten raw? Do you have in mind any kind of fermented sausages intended to fry/cook?

Response to question 1: Thanks for your suggestion. The fermented sausage made in this study was mainly used to eat raw. Therefore, the effect of cooking and frying on biogenic amine formation of the fermented sausage was out of the point of this work. However, your suggestion is another good point, we will take consideration in our further experimental study. We hope you could pay attention to our follow-up researches continuously.

Question 2: Line 73 the reference number is missing at Skaliac et al.

Response to question 2: Sorry for this funny mistake. We have corrected the mistake in our revised manuscript (line 89, page 5).

Question 3: Line 74 US-EPA PAHs should be explained.

Response to question 3: Thank you for your careful review. We have added relevant content in our revised manuscript (lines 87-88, page 5).

Question 4: Line 75 P<0.05 P should be in italics.

Response to question 4: Sorry for this mistake. We have deleted this part.

Question 5: Line 76 It should be Zajac et al not Marzena and the number should be at the names as in like 73.

Response to question 5: Sorry for this mistake. We have corrected the mistake in our revised manuscript (line 89, page 5).

Question 6: Line 125 the space is missing between pH and 4.0

Response to question 6: Sorry for this mistake and thanks for your careful review. We have corrected the mistake in our revised manuscript (line 153, page 8).

Question 7: Line 128 The microbiological analysis should be properly described: type of diluting solution, names of microbiological media used, conditions of incubations (time/temperature).

Response to question 7: Thank you for your careful review. We have revised some contents in our revision (lines 157-166, pages 8-9).

Question 8: Line 135 the type of texture meter should be mentioned.

Response to question 8: Thank you for your careful review. We have revised some contents in our revision (lines 169-170, page 9).

Question 9: Line 138 1 cm×1 cm3 3 is missing at first cm.

Response to question 9: Sorry for this mistake. We have corrected the mistake in our revised manuscript (line 173, page 9).

Question 10: Line 139 the compression conditions should be included.

Response to question 10: Thank you for your careful review. We have revised some contents in our revision (lines 170-172, page 9).

Question 11: Line 141. The type of chromatograph and detector used should be included. The methods could be described in more detail.

Response to question 11: Thanks for your suggestion. We have added some contents in our revision (lines 180-187, pages 9-10).

Question 12: Table 1 the significance letters should be at mean values not at errors.

Response to question 12: Thank you for your careful review. We think adding standard errors makes the data more reasonable for our results.

Question 13: Line 170 The information what PC or CC stand for should be added – the table should be self-explanatory.

Response to question 13: Thank you for your careful review. CC group and PC group respectively represented collagen casings and pig casings group, which we have mentioned at materials and methods part and we have also added this content in our revision as your suggestion. (lines 219, page 12).

Question 14: Line 183-190 this information was already given in Materials and methods section.

Response to question 14: Thank you for your careful review. We have deleted this part in our revision.

Question 15: Line 202 There were significant differences between different capital letters in the same group at different periods (P<0.05). There were significant differences in different lowercase letters among different groups in the same period (P<0.05). – it should be shortened eg. Capital letters indicate….

Response to question 15: Thank you for your careful review. We have revised these in Table.1, Table.2, Table.3, Table.4, Table.5, Table.6, of our revised manuscript.

Question 16: Line 234 There were several studies in which fermented sausages produced in collagen casings were produced. Was the same low pH in CC detected? It would be worth discussing.

Response to question 16: Thanks for your suggestion. We have added some contents in our revision (lines 281-283, page 17-18).

Question 17: Line 240 This comment would be adequate if the casings production would be a subject of the study. In my opinion it is not suitable here.

Response to question 17: Thank you for your careful review. We have revised this part in our revised manuscript (lines 327-329, page 18-19).

Question 18: Line 243 It should be paraphrased. Tenderness is one of the texture parameters – there is nothing to believe in.

Response to question 18: Thank you for your careful review. We have corrected this part in our revised manuscript (line 290, page 17).

Question 19: Line 245 Why was it improved? Is harder better?

Response to question 19: Thanks for your suggestion. We apologized for our written organization skills. The hardness of fermented sausage in CC group was larger than that in PC group, which made fermented sausage more compact. Mei Qin Feng, Jie Zhang & Jian Sun. Effects of imitated Staphylococcus inoculation on quality and oxidation stability of fermented sausage. Food Science, 1-13. doi:10.7506/ SPKX1002-6630-20210805-062. (in Chinese).

---

## [Decision Letter · Decision Letter 1]

1 Dec 2021

PONE-D-21-28210R1Effect of Collagen Casing on the Quality Characteristics of Fermented SausagePLOS ONE

Dear Dr. Duan,

Thank you for submitting your manuscript to PLOS ONE. After careful consideration, we feel that it has merit but does not fully meet PLOS ONE’s publication criteria as it currently stands. Therefore, we invite you to submit a revised version of the manuscript that addresses the points raised during the review process.

We look forward to receiving your revised manuscript.

Kind regards,

A. M. Abd El-Aty

Academic Editor

PLOS ONE

AE Comments:

Still, reviewer # 3 is raising some concern over the revised form of the MS. Do go through the comments and amend the MS accordingly. Further,<o:p></o:p>

1- The P letter for statistical value should be an uppercase-italic face letter. Amend throughout the text<o:p></o:p>

2- Line 33: The results showed that with storage time increasing...amend to The results showed that with increasing the storage time, <o:p></o:p>

3- Lines 81-91:Raise a question about the novel aspect of the current study. If there were several studies (2,5, 13, 14, 15), then "what does the current MS add to the existing knowledge"? This Must be discussed and interpreted in the introduction. <o:p></o:p>

4- Full vendor details should include company, city (state) and country. Amend throughout the text<o:p></o:p>

5- Use mL, uL, L, dL throughout the MS<o:p></o:p>

6- Check spaces and punctuation throughout the MS<o:p></o:p>

7- Section: Determination of biogenic amines: You should provide the details of method performance. This should include calibration curves, linearity, LOD, LOQ, accuracy and precision of quantified biogenic amine <o:p></o:p>

8- Data in Tables should be presented as mean +/-SD, not SE<o:p></o:p>

9- Conclusion: Rewrite it again as it repeats what has been stated in the abstract<o:p></o:p>

Reviewers' comments:

Reviewer's Responses to Questions

**Comments to the Author**

1. If the authors have adequately addressed your comments raised in a previous round of review and you feel that this manuscript is now acceptable for publication, you may indicate that here to bypass the “Comments to the Author” section, enter your conflict of interest statement in the “Confidential to Editor” section, and submit your "Accept" recommendation.

Reviewer #1: All comments have been addressed

Reviewer #3: All comments have been addressed

Reviewer #5: All comments have been addressed

2. Is the manuscript technically sound, and do the data support the conclusions?

Reviewer #1: Yes

Reviewer #3: Partly

Reviewer #5: Yes

3. Has the statistical analysis been performed appropriately and rigorously? 

Reviewer #1: Yes

Reviewer #3: Yes

Reviewer #5: Yes

4. Have the authors made all data underlying the findings in their manuscript fully available?

Reviewer #1: Yes

Reviewer #3: Yes

Reviewer #5: Yes

5. Is the manuscript presented in an intelligible fashion and written in standard English?

Reviewer #1: Yes

Reviewer #3: Yes

Reviewer #5: Yes

6. Review Comments to the Author

Reviewer #1: Dear Editor,

The authors have revised their manuscript according to the reviewers' suggestions. Therefore, it can be accepted in its current form.

Reviewer #3: I think the sample size is too limited, especially for the texture analysis. Also the speed of measuring with TPA is rather low.

The story would have been much stronger when microscopy analysis and permeability of the casings was done.

Reviewer #5: I have only one comment regarding the indication of statistical significance. WHat I mean was that the letters should be at mean values and not at errors. e.g. 20a +/- 0.1 and not 20 +/- 0.1a . Using standard errors was correct of course.

7. PLOS authors have the option to publish the peer review history of their article (what does this mean?). If published, this will include your full peer review and any attached files.

Reviewer #1: No

Reviewer #3: No

Reviewer #5: No

---

## [Author Response · Author response to Decision Letter 1]

3 Dec 2021

Dear editor:

Thank you for providing these comments on our submission entitled “Effect of Collagen Casing on the Quality Characteristics of Fermented Sausage”. We appreciate the constructive comments from you and five reviewers. We have studied the comments carefully and provided a detailed response to each point raised.

Response to Reviewer 

Question 1: The P letter for statistical value should be an uppercase-italic face letter. Amend throughout the text.

Response to question 1: Thanks for your careful review. We have corrected the mistake in our revised manuscript (line 35, 36, 38, page 2).

Question 2: Line 33: The results showed that with storage time increasing...amend to The results showed that with increasing the storage time.

Response to question 2: Thanks for your suggestion. We have corrected relevant content in our revised manuscript (lines 31, page 2).

Question 3: Lines 81-91: Raise a question about the novel aspect of the current study. If there were several studies (2,5, 13, 14, 15), then "what does the current MS add to the existing knowledge"? This Must be discussed and interpreted in the introduction.

Response to question 3: Thanks for your suggestion. We have added relevant content in our revised manuscript (lines 95-96, pages 5-6).

Question 4: Full vendor details should include company, city (state) and country. Amend throughout the text.

Response to question 4: Sorry for this mistake. We have corrected the mistake in our revised manuscript (line 105-108, 118, 120, 123, 137, 148-149, 155, 161,163, 165,182 pages 6-9).

Question 5: Use mL, uL, L, dL throughout the MS.

Response to question 5: Sorry for this mistake. We have corrected the mistake in our revised manuscript (line 150 page 8).

Question 6: Check spaces and punctuation throughout the MS.

Response to question 6: Thank you for your constructive suggestion. We have added relevant content in our revised manuscript (lines 105, 159, pages 6, 8).

Question 7: Section: Determination of biogenic amines: You should provide the details of method performance. This should include calibration curves, linearity, LOD, LOQ, accuracy and precision of quantified biogenic amine.

Response to question 7: Thank you for your constructive suggestion. We have added relevant content in our revised manuscript (lines 191-195, page 10).

Question 8: Data in Tables should be presented as mean +/-SD, not SE.

Response to question 8: Sorry for this funny mistake. Actually we have presented our data as mean±SD. We have modified mean±SE to mean±SD in our revised manuscript (lines 209, 224, 237, 243, 257, 274 pages 11-14, 16). If you have any doubts, please check our original data uploaded before. 

Question 9: Conclusion: Rewrite it again as it repeats what has been stated in the abstract.

Response to question 9: Thank you for your suggestion. We have added relevant content in our revised manuscript (lines 335-337, page 19).

---

## [Editor Report · Decision Letter 2]

9 Dec 2021

PONE-D-21-28210R2

Effect of Collagen Casing on the Quality Characteristics of Fermented Sausage

PLOS ONE

Dear Dr. Duan,

Thank you for submitting your manuscript to PLOS ONE. After careful consideration, we feel that it has merit but does not fully meet PLOS ONE’s publication criteria as it currently stands. Therefore, we invite you to submit a revised version of the manuscript that addresses the points raised during the review process.

ACADEMIC EDITOR: So many files were uploaded. Upload only the last revised MS with response to the Editor's/reviewers comments. All corrections should be highlighted yellow (not in track-changes mode).

We look forward to receiving your revised manuscript.

Kind regards,

A. M. Abd El-Aty

Academic Editor

PLOS ONE
---

## [Author Response · Author response to Decision Letter 2]

10 Dec 2021

Dear editor:

Dear editor:

Thank you for providing these comments on our submission entitled “Effect of Collagen Casing on the Quality Characteristics of Fermented Sausage”. We appreciate the constructive comments from you and five reviewers. We have studied the comments carefully and provided a detailed response to each point raised.

Response to Reviewer 

Question 1: The P letter for statistical value should be an uppercase-italic face letter. Amend throughout the text.

Response to question 1: Thanks for your careful review. We have corrected the mistake in our revised manuscript (line 35, 36, 38, page 2).

Question 2: Line 33: The results showed that with storage time increasing...amend to The results showed that with increasing the storage time.

Response to question 2: Thanks for your suggestion. We have corrected relevant content in our revised manuscript (lines 30, 31, page 2).

Question 3: Lines 81-91: Raise a question about the novel aspect of the current study. If there were several studies (2,5, 13, 14, 15), then "what does the current MS add to the existing knowledge"? This Must be discussed and interpreted in the introduction.

Response to question 3: Thanks for your suggestion. We have added relevant content in our revised manuscript (lines 94-96, pages 5-6).

Question 4: Full vendor details should include company, city (state) and country. Amend throughout the text.

Response to question 4: Sorry for this mistake. We have corrected the mistake in our revised manuscript (line 104-107, 117, 119, 122, 136, 147-148, 154, 160, 162, 164,181 pages 6-9).

Question 5: Use mL, uL, L, dL throughout the MS.

Response to question 5: Sorry for this mistake. We have corrected the mistake in our revised manuscript (line 149, 158 page 8).

Question 6: Check spaces and punctuation throughout the MS.

Response to question 6: Thank you for your constructive suggestion. We have added relevant content in our revised manuscript (lines 105, 159, pages 6, 8).

Question 7: Section: Determination of biogenic amines: You should provide the details of method performance. This should include calibration curves, linearity, LOD, LOQ, accuracy and precision of quantified biogenic amine.

Response to question 7: Thank you for your constructive suggestion. We have added relevant content in our revised manuscript (lines 190-194, page 10).

Question 8: Data in Tables should be presented as mean +/-SD, not SE.

Response to question 8: Sorry for this funny mistake. Actually we have presented our data as mean±SD. We have modified mean±SE to mean±SD in our revised manuscript (lines 208, 235, 248, 254, 280, 297 pages 11-14, 16). If you have any doubts, please check our original data uploaded before. 

Question 9: Conclusion: Rewrite it again as it repeats what has been stated in the abstract.

Response to question 9: Thank you for your suggestion. We have added relevant content in our revised manuscript (lines 358-360, page 19).

---

## [Decision Letter · Decision Letter 3]

17 Dec 2021

PONE-D-21-28210R3Effect of Collagen Casing on the Quality Characteristics of Fermented SausagePLOS ONE

Dear Dr. Duan,

Thank you for submitting your manuscript to PLOS ONE. After careful consideration, we feel that it has merit but does not fully meet PLOS ONE’s publication criteria as it currently stands. Therefore, we invite you to submit a revised version of the manuscript that addresses the points raised during the review process.

ACADEMIC EDITOR: The authors have to provide the additional data requested by the diligent reviewer. This will be the last option for revision. Without doing this, the MS will not considered for publication in PLOS ONE.

We look forward to receiving your revised manuscript.

Kind regards,

A. M. Abd El-Aty

Academic Editor

PLOS ONE

Reviewers' comments:

Reviewer's Responses to Questions

**Comments to the Author**

1. If the authors have adequately addressed your comments raised in a previous round of review and you feel that this manuscript is now acceptable for publication, you may indicate that here to bypass the “Comments to the Author” section, enter your conflict of interest statement in the “Confidential to Editor” section, and submit your "Accept" recommendation.

Reviewer #2: (No Response)

Reviewer #3: All comments have been addressed

2. Is the manuscript technically sound, and do the data support the conclusions?

Reviewer #2: Partly

Reviewer #3: Yes

3. Has the statistical analysis been performed appropriately and rigorously? 

Reviewer #2: Yes

Reviewer #3: Yes

4. Have the authors made all data underlying the findings in their manuscript fully available?

Reviewer #2: No

Reviewer #3: Yes

5. Is the manuscript presented in an intelligible fashion and written in standard English?

Reviewer #2: Yes

Reviewer #3: Yes

6. Review Comments to the Author

Reviewer #2: Although the authors improved the quality of the manuscript, they did not incorporate the additional data recommended during the past reviews and the manuscript is still superficial. The authors were advised to add data related to the variations of fats, fatty acid, and other common volatile components, but failed to do so claiming they will consider such results for another publication. Besides, results from optical microscope (SEM) analysis are still lacking. With a focus on some biogenic amines and a few physical characteristics only, the manuscript is not in the required standard to be considered for publication.

Reviewer #3: The authors have incorporated the comments. I understand that running extra texture measurements and incorporate permeability measurements is not possible. Maybe a suggestion for future articles. This article is now ready for publishing.

7. PLOS authors have the option to publish the peer review history of their article (what does this mean?). If published, this will include your full peer review and any attached files.

Reviewer #2: No

Reviewer #3: No

---

## [Author Response · Author response to Decision Letter 3]

31 Dec 2021

Dear editor:

Thank you for providing these comments on our submission entitled “Effect of Collagen Casing on the Quality Characteristics of Fermented Sausage”. We appreciate the constructive comments from you and five reviewers. We have studied the comments carefully and provided a detailed response to each point raised.

Response to Reviewer 2 

Question 1: The manuscript must describe a technically sound piece of scientific research with data that supports the conclusions. Experiments must have been conducted rigorously, with appropriate controls, replication, and sample sizes. The conclusions must be drawn appropriately based on the data presented.

Response to question 1: We have checked our experiment design and original data, meanwhile have added some experiments on volatile components to make our results experiments more complete.

Question 2: Although the authors improved the quality of the manuscript, they did not incorporate the additional data recommended during the past reviews and the manuscript is still superficial. The authors were advised to add data related to the variations of fats, fatty acid, and other common volatile components, but failed to do so claiming they will consider such results for another publication. Besides, results from optical microscope (SEM) analysis are still lacking. With a focus on some biogenic amines and a few physical characteristics only, the manuscript is not in the required standard to be considered for publication.

Response to question 2: Thank you for your suggestion. we have measured the fermented sausage flavor and showed relevant contents in our revised manuscript (line 36-37, page 2, line 180-184, page 9, 269-286, 15-18, 336-349, page 21-22 and table 6). Our research mainly focused on the effect of different casing types on the quality characteristics of fermented sausage. The research on the organizational structure of casings itself was not the focus of this research. Your valuable suggestions have been involved in our subsequent research. Please keep an eye on our research progress.

Response to Reviewer 3 

Question 1: The authors have incorporated the comments. I understand that running extra texture measurements and incorporate permeability measurements is not possible. Maybe a suggestion for future articles. This article is now ready for publishing.

Response to question 1: Thank you very much for your valuable suggestions. We will adopt the measurement of relevant indicators you proposed in the following research. Please keep paying attention to us.

---

## [Decision Letter · Decision Letter 4]

5 Jan 2022

PONE-D-21-28210R4Effect of Collagen Casing on the Quality Characteristics of Fermented SausagePLOS ONE

Dear Dr. Duan,

Thank you for submitting your manuscript to PLOS ONE. After careful consideration, we feel that it has merit but does not fully meet PLOS ONE’s publication criteria as it currently stands. Therefore, we invite you to submit a revised version of the manuscript that addresses the points raised during the review process.

ACADEMIC EDITOR: Further justifications are needed as raised by the diligent reviewer. Do go through the comments and amend the MS accordingly. Proofread the text for grammar and syntax errors, if any.

We look forward to receiving your revised manuscript.

Kind regards,

A. M. Abd El-Aty

Academic Editor

PLOS ONE

Reviewers' comments:

Reviewer's Responses to Questions

**Comments to the Author**

1. If the authors have adequately addressed your comments raised in a previous round of review and you feel that this manuscript is now acceptable for publication, you may indicate that here to bypass the “Comments to the Author” section, enter your conflict of interest statement in the “Confidential to Editor” section, and submit your "Accept" recommendation.

Reviewer #2: (No Response)

2. Is the manuscript technically sound, and do the data support the conclusions?

Reviewer #2: Partly

3. Has the statistical analysis been performed appropriately and rigorously? 

Reviewer #2: Yes

4. Have the authors made all data underlying the findings in their manuscript fully available?

Reviewer #2: Yes

5. Is the manuscript presented in an intelligible fashion and written in standard English?

Reviewer #2: No

6. Review Comments to the Author

Reviewer #2: With the addition of data related to volatile components, the content of the manuscript has now improved to some extent. Related to this, the authors should clarify the method regarding the analysis of volatile components (lines 180-184). Specifically, the modified method, the identification procedure, and the quantification techniques along with key GC-MS parameters should be described. More importantly, it would be great if the authors incorporate the effect of storage time (21 d, 41 d and 61d) on the variations of these volatile molecules (as they did for biogenic amines).

7. PLOS authors have the option to publish the peer review history of their article (what does this mean?). If published, this will include your full peer review and any attached files.

Reviewer #2: No

---

## [Author Response · Author response to Decision Letter 4]

12 Jan 2022

Dear editor:

Thank you for providing these comments on our submission entitled “Effect of Collagen Casing on the Quality Characteristics of Fermented Sausage”. We appreciate the constructive comments from you and five reviewers. We have studied the comments carefully and provided a detailed response to each point raised.

Response to Reviewer 2 

Question 1: ThIs the manuscript technically sound, and do the data support the conclusions? The manuscript must describe a technically sound piece of scientific research with data that supports the conclusions. Experiments must have been conducted rigorously, with appropriate controls, replication, and sample sizes. The conclusions must be drawn appropriately based on the data presented.

Response to question 1: We have checked our experiment design and original data, meanwhile have added some experiments on volatile components to make our results experiments more complete.

Question 2: Is the manuscript presented in an intelligible fashion and written in standard English?

Response to question 2: Thanks for your careful check. We sought for the help from AJE (American Journal Experts) to refine our manuscript (D3C61K32).

Response to question 2: 

Question 3: With the addition of data related to volatile components, the content of the manuscript has now improved to some extent. Related to this, the authors should clarify the method regarding the analysis of volatile components (lines 180-184). Specifically, the modified method, the identification procedure, and the quantification techniques along with key GC-MS parameters should be described. More importantly, it would be great if the authors incorporate the effect of storage time (21 d, 41 d and 61d) on the variations of these volatile molecules (as they did for biogenic amines).

Response to question 3: Thank you for your suggestion. We have added relevant contents such as the determination method, results and discussion of fermented sausage flavor. (line36-37, 48, 185-189, 273-285, 374-375, page 2, 3, 9-10, 15-20, 25).

---

## [Decision Letter · Decision Letter 5]

19 Jan 2022

Effect of Collagen Casing on the Quality Characteristics of Fermented Sausage

PONE-D-21-28210R5

Dear Dr. Duan,

We’re pleased to inform you that your manuscript has been judged scientifically suitable for publication and will be formally accepted for publication once it meets all outstanding technical requirements.

Kind regards,

A. M. Abd El-Aty

Academic Editor

PLOS ONE

Additional Editor Comments (optional):

Reviewers' comments:

Reviewer's Responses to Questions

**Comments to the Author**

1. If the authors have adequately addressed your comments raised in a previous round of review and you feel that this manuscript is now acceptable for publication, you may indicate that here to bypass the “Comments to the Author” section, enter your conflict of interest statement in the “Confidential to Editor” section, and submit your "Accept" recommendation.

Reviewer #2: All comments have been addressed

2. Is the manuscript technically sound, and do the data support the conclusions?

Reviewer #2: Yes

3. Has the statistical analysis been performed appropriately and rigorously? 

Reviewer #2: Yes

4. Have the authors made all data underlying the findings in their manuscript fully available?

Reviewer #2: Yes

5. Is the manuscript presented in an intelligible fashion and written in standard English?

Reviewer #2: No

6. Review Comments to the Author

Reviewer #2: The authors have addressed the issues raised during the past review processes.

The manuscript is now in a good standard and can be considered for publication after rigorous language edition.

7. PLOS authors have the option to publish the peer review history of their article (what does this mean?). If published, this will include your full peer review and any attached files.

Reviewer #2: No

---

## [Editor Report · Acceptance letter]

25 Jan 2022

PONE-D-21-28210R5 

Effect of collagen casing on the quality characteristics of fermented sausage 

Dear Dr. Duan:

I'm pleased to inform you that your manuscript has been deemed suitable for publication in PLOS ONE. Congratulations! Your manuscript is now with our production department. 

Kind regards, 

on behalf of

Prof. A. M. Abd El-Aty 

Academic Editor

PLOS ONE